# Foxp3 orchestrates reorganization of chromatin architecture to establish regulatory T cell identity

Zhi Liu[1,2,5], Dong-Sung Lee [3,4,5], Yuqiong Liang[2], Ye Zheng [2,6] & Jesse R. Dixon [3,6]

Chromatin conformation reorganization is emerging as an important layer of regulation for gene expression and lineage specification. Yet, how lineage-specific transcription factors contribute to the establishment of cell type-specific 3D chromatin architecture in the immune cells remains unclear, especially for the late stages of T cell subset differentiation and maturation. Regulatory T cells (Treg) are mainly generated in the thymus as a sub-population of T cells specializing in suppressing excessive immune responses. Here, by comprehensively mapping 3D chromatin organization during Treg cell differentiation, we show that Treg-specific chromatin structures were progressively established during its lineage specification, and highly associated with Treg signature gene expression. Additionally, the binding sites of Foxp3, a Treg lineage specifying transcription factor, were highly enriched at Treg-specific chromatin loop anchors. Further comparison of the chromatin interactions between wide-type Tregs versus Treg cells from Foxp3 knock-in/knockout or newly-generated Foxp3 domain-swap mutant mouse revealed that Foxp3 was essential for the establishment of Treg-specific 3D chromatin architecture, although it was not dependent on the formation of the Foxp3 domain-swapped dimer. These results highlighted an underappreciated role of Foxp3 in modulating Treg-specific 3D chromatin structure formation.

Genome-wide 3D chromosome conformation capture technologies have revealed that higher-order 3D chromatin structures of mammalian genome are hierarchically organized into chromosome territories, A/B compartments, topologically associating domains (TADs), and chromatin loops[1–5]. TADs are genomic regions that self-interact but insulate regions outside the domain, therefore contributing to the regulation of gene expression by restricting interactions of cis-regulatory elements to their target genes[6–8]. The zinc-finger transcription factor (TF) CTCF and the ring-shaped cohesin complex play critical roles in the formation and maintenance of TADs across different cell lineages[9]. However, these ubiquitously expressed proteins alone cannot establish and maintain cell lineage-specific genome architecture. Lineage-specific TFs have been proposed to regulate genome organization in specific cell lineages[8,10,11]. As for T cells in the immune system, it has been revealed that Bcl11b and TCF1 controls 3D chromatin architectures during early T cell development[12–14]. However, little is known about global genome organization for the late stage of T cell development and differentiation.

T cell development in the thymus occurs through several stages distinguished by the expression of cell surface markers CD4 and CD8.

[1]Shanghai Immune Therapy Institute, Renji Hospital, Shanghai Jiao Tong University School of Medicine, Shanghai, China. [2]NOMIS Center for Immunobiology and Microbial Pathogenesis, Salk Institute for Biological Studies, La Jolla, CA, USA. [3]Gene Expression Laboratory, Salk Institute for Biological Studies, La Jolla, CA, USA. [4]Department of Life Sciences, University of Seoul, Seoul, South Korea. [5]These authors contributed equally: Zhi Liu, Dong-Sung Lee. [6]These authors jointly supervised this work: Ye Zheng, Jesse R. Dixon. ✉e-mail: yzheng@salk.edu; jedixon@salk.edu

Early T cell precursors are CD4/CD8 double-negative (DN), which differentiate into the CD4/CD8 double-positive (DP) intermediate population, and then the mature CD4 single-positive (SP) or CD8 SP populations[12,15] (Fig. 1a). Regulatory T cells (Treg) are derived in the thymus from CD4 SP cells, through two distinct developmental programs involving either CD25+Foxp3− or CD25−Foxp3+ Treg cell precursors, both of which develop into CD25+Foxp3+ mature Treg cells to maintain immune tolerance and homeostasis[16–18]. Treg cell lineage specification represents one of the final stages of T cell development and thus is an excellent model to examine the roles of lineage-specific TFs in 3D genome organization[18,19]. Foxp3, an X chromosome-encoded gene in the forkhead TF family, plays a central role in Treg cell lineage specification, phenotypic stability, metabolic fitness, and regulatory function[20–27]. Depending on the activating or repressing cofactors it associates with[24,28], Foxp3 can either promote or inhibit target gene expression by exploiting pre-existing enhancer landscapes[29]. A recent study using HiChIP showed that Foxp3 was associated with enhancer-promoter loops to fine tune Foxp3-dependent gene expression[30]. However, it remains elusive how the Treg 3D chromatin architecture is established during their lineage specification, how it influences gene expression in Tregs, and whether and how Foxp3 contributes to Treg-specific chromatin interactions. Considering it was reported that Foxp3 has the capability to bring two distal DNA elements together through the formation a domain-swapped dimer[31], it would be of great interest to test whether that Foxp3 can function as a loop anchor protein to directly contribute to the establishment of Treg-specific TADs and chromatin loops.

Here, we comprehensively mapped the chromatin interactions across different Treg developmental stages in the thymus, and Treg and Tcon cells in the spleen by in situ Hi-C. Our findings revealed that the 3D genome of Treg cells is gradually established during Treg cell development, and Treg-specific chromatin interactions were associated with Treg signature gene expression. Furthermore, through comparison of WT Treg cells with "wannabe" Treg cells (isolated from Foxp3-GFP knock-in/knockout mouse), and Foxp3 domain-swap mutant (DSM) Treg cells (isolated from a newly-generated Foxp3 DSM mouse strain), our data showed that Foxp3 was critical for the establishment of Treg-specific chromatin interactions, although not likely dependent on the Foxp3 domain-swapped dimer to form Foxp3-associated chromatin loops. Our results revealed a previously unappreciated aspect of Foxp3 function in the regulation of Treg cell development and function.

## Results

### Global transformation of 3D genome architecture during Treg lineage specification

To map the trajectory of the 3D genome organization during Treg lineage development, we performed in situ Hi-C experiments with T cells in different developmental stages, including CD4−CD8− (DN), CD4+CD8+ (DP), CD4−CD8+ (CD8SP), CD4+CD8−Foxp3−CD25− (CD4SP), CD4+CD8−Foxp3−CD25+ (CD25+ Treg precursor), CD4+CD8−Foxp3loCD25− (Foxp3lo Treg precursor), CD4+CD8−Foxp3+CD25+ (Treg) from the thymus, and CD4+Foxp3− (conventional T cells, Tcon) and CD4+Foxp3+ (mature Treg) from the spleen (Fig. 1a, Supplementary Fig. 1a, 1b, and Supplementary Data 1). In situ Hi-C is a genome-wide variant of chromatin conformation assay that provides an all-to-all, high resolution view of chromatin interactions in the genome[2]. Chromatin interaction maps were constructed from in situ Hi-C data at different resolutions down to 2 kb resolution (Fig. 1b). To examine patterns of gains and losses of chromatin interactions, we performed K-means clustering on normalized Hi-C contacts across each T cell population at 100 kb resolution (Fig. 1c). This revealed evidence for both gains and losses of chromatin interactions throughout the T cell developmental trajectory (Fig. 1c, d). We next use T-distributed Stochastic Neighbor Embedding

(t-SNE) map to visualize and compare different T cell subsets in two dimensions. t-SNE showed tight grouping between replicate experiments from the same T cell subset, indicating high reproducibility of the chromatin contact maps. Furthermore, the t-SNE map clearly showed a differentiation trajectory of Treg cell development in the thymus, in parallel to the trajectory generated based on gene expression profile of each T cell subset (Fig. 1e). Therefore, the 3D genome structure is being reorganized following the differential steps of thymic Treg cell development.

**Treg-specific chromatin contacts are associated with Treg gene expression and selective transcription factor binding activities**
To examine whether Treg cells have a unique 3D chromatin structure to support their distinct gene expression profile and function, we performed a pairwise comparison of the 3D chromatin structure of splenic Treg and Tcon cells. Using edgeR to detect differential chromatin contacts at 25 kB resolution, we identified 1959 upregulated DNA interactions in Treg cells and 1973 upregulated DNA interactions in Tcon cells (FDR 1%, Fig. 2a, Supplementary Data 2). As an example, we observed strong DNA interactions close to the *Socs2* gene locus in Treg cells, but not in conventional T cells (Fig. 2b). The Hi-C data showed that a vast majority of DNA interactions were similar between Treg and Tcon, with only 0.29% of tested chromatin interactions showing a significant change between Treg and Tcon cells (3932/1377570). This small percentage agreed with previous studies showing that less than 5% of the genes are differentially expressed in these two T cell populations[22,32,33]. Additionally, Treg and Tcon cells share 99% of their enhancers, less than 1% of the enhancers are unique to either T cell population[29]. In examining the changes in Treg and Tcon specific chromatin interactions during Treg lineage commitment, we observe that Treg specific interactions most dramatically differ upon expression of Foxp3 and some are further strengthened in Foxp3+ CD25+ cells (Supplementary Fig. 1c). In contrast, most Tcon specific interactions are lost upon expression of Foxp3 and do not show further reductions in Foxp3+ CD25+ cells (Supplementary Fig. 1c), suggesting that Foxp3 expression is the major distinguishing characteristic of when Treg versus Tcon specific chromatin interactions emerge.

We next examined whether the differences in 3D chromatin structures between Treg and Tcon cells are associated with changes in gene expression. Analysis of gene expression profiles of Treg and Tcon cells revealed that 928 genes (FDR < 1% and Fold change >2) were differentially expressed (Fig. 2c). Based on Hi-C data, of the 3932 differential chromatin interactions between Treg and Tcon cells, 617 chromatin interaction anchors overlapped with the transcription start site of at least one gene. Strikingly, differentially expressed genes were strongly enriched at differential chromatin interactions, with Treg specific genes enriched at Treg specific chromatin interactions, and Tcon specific genes enriched at Tcon specific chromatin interactions (Fig. 2d). This result suggested that differences in gene expression are potentially associated with unique DNA interacting activities in Treg cells.

To further explore the link between chromatin interactions and gene expression, we determined the number of differential interacting partners for each individual interaction anchor that is proximal to a differentially expressed gene. While most loci with differential interactions had only one partner anchor, a minority of loci form multiple differential long-range contacts (Fig. 2e). We observed that the loci with higher numbers of differential long-range interactions were more likely to be associated with differentially expressed genes (Fig. 2f, g). Of note, the loci with more than four differential DNA interactions were associated with several well-established Treg signature genes, including *Ikzf2*, *Lrrc32* (encoding GARP), *Socs2*, and *Ptger4* (Fig. 2g, h). These results further illustrated the correlation between changes in DNA interactions and differential gene expression in Treg cells.

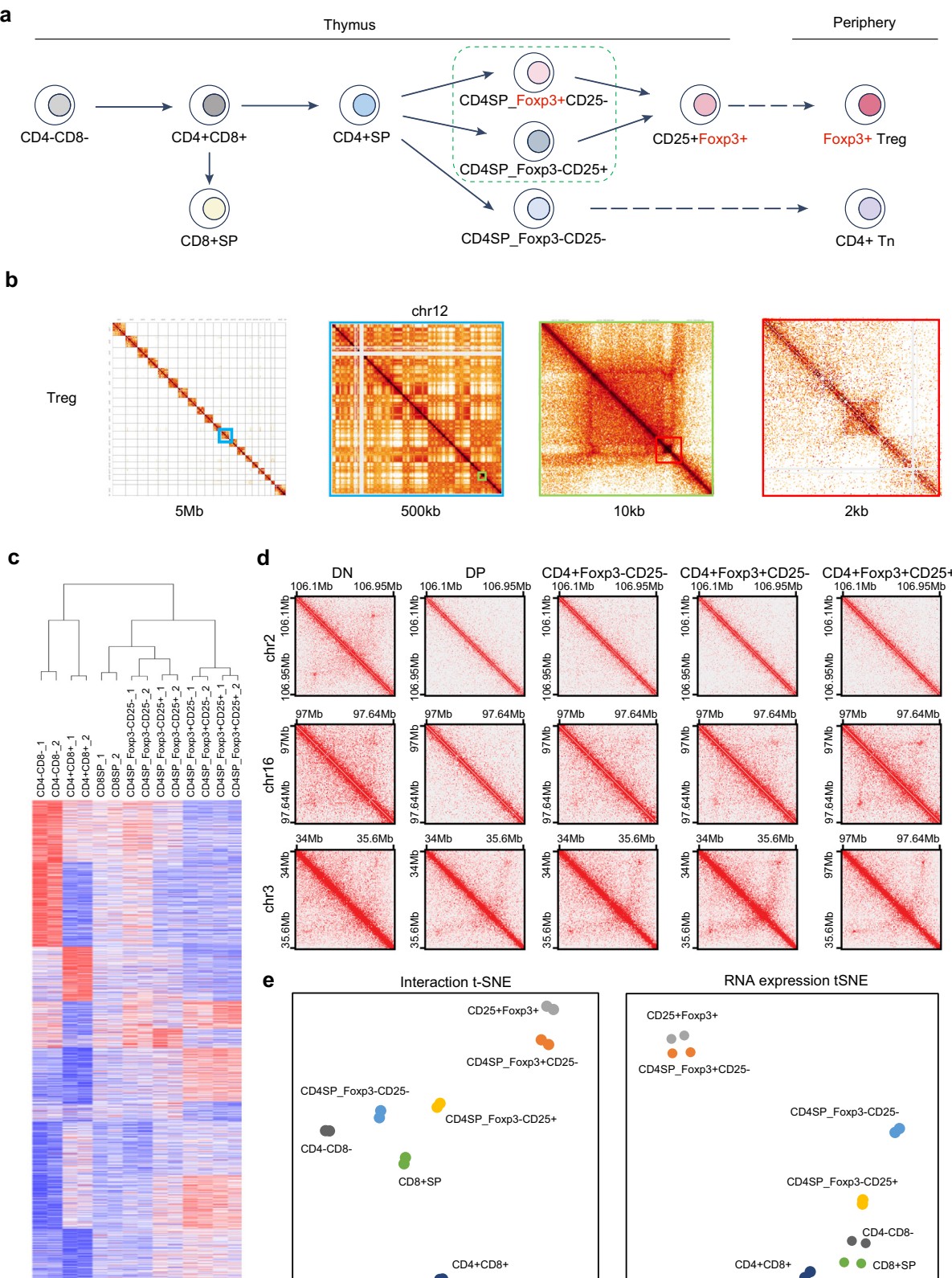

Fig. 1 | Chromatin architecture reorganization during T cell lineage commitment. a Schematic of experimental design to study changes in chromatin architecture during T cell lineage commitment. b Example Hi-C heat maps in mature splenic Treg cells at progressively higher resolutions near the Bcl11b gene. c Clustering of chromatin interactions in thymic T cell subsets. Chromatin interactions were calculated at a resolution of 100 kb and clustered using K-means clustering (K = 10). Replicate experiments were also clustered using hierarchical clustering. d Hi-C chromatin interactions over specific loci that gain or lose chromatin contacts during T cell lineage commitment. e T-distributed stochastic neighbor embedding (tSNE) of chromatin contacts (left panel) and gene expression (right panel) during T cell lineage commitment.

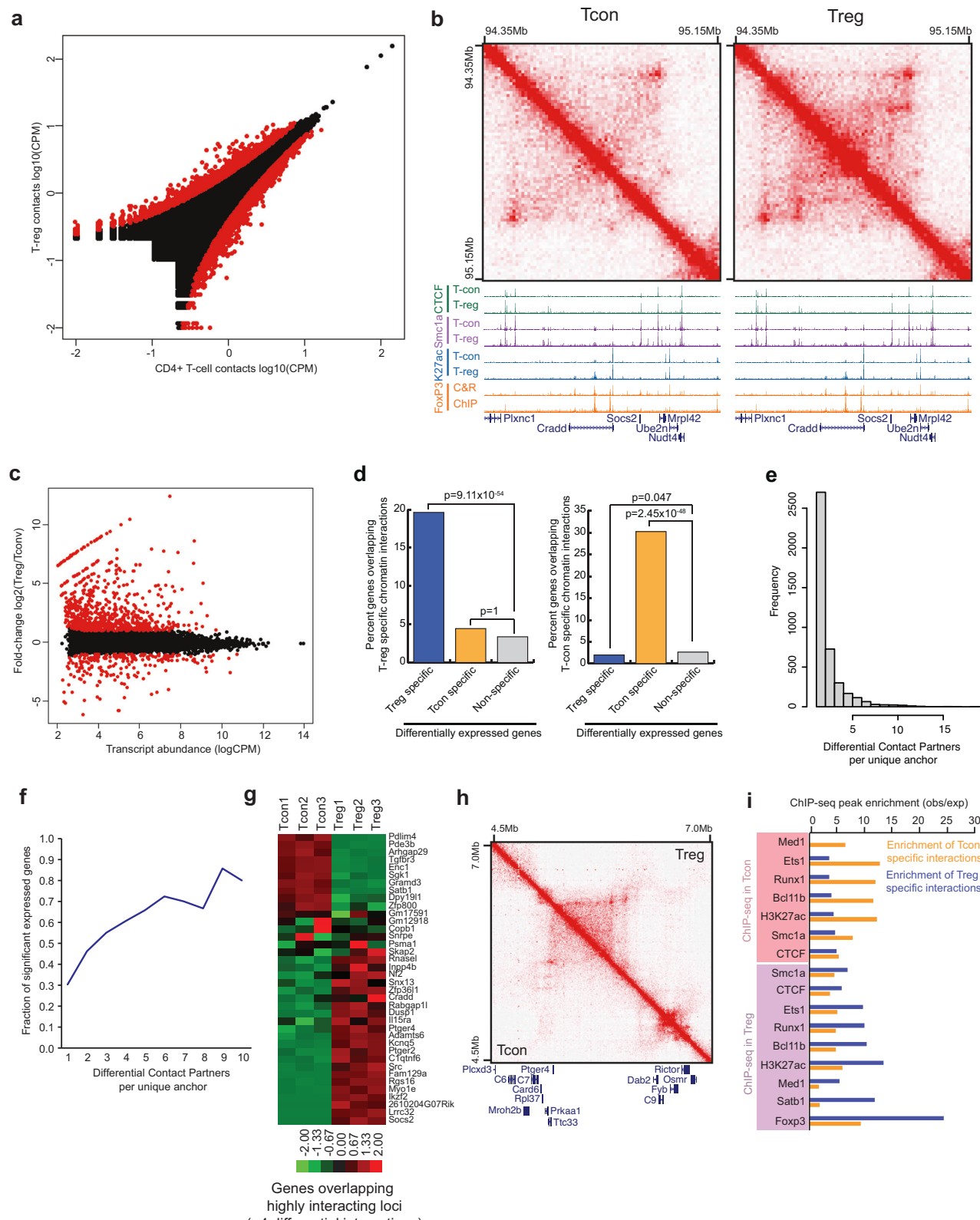

To search for factors involved in establishing a Treg-specific 3D chromatin structure, we compared the relative enrichment of several TFs and chromatin structure regulators in Treg- and Tcon- specific loop anchors. Although CTCF and cohesin are critical for the formation of DNA loops[34,35], their relative enrichments in Tcon or Treg-specific loop anchors were moderate (Fig. 2i). In contrast, multiple TFs, including Foxp3, Satb1, Runx1, Ets1, and Bcl11b, which are involved in Treg differentiation and function, showed a more pronounced enrichment in Tcon or Treg-specific interactions. Strikingly, the enrichment of Foxp3 bound peaks was ranked first among all the TFs tested, suggesting that Foxp3 might play a crucial role in Treg-specific loop formation (Fig. 2i).

**Fig. 2 | Chromatin conformation changes between Treg cells and conventional CD4⁺ T cells. a** Chromatin interaction frequency of 25 kb bins compared between conventional CD4⁺ T cells (x-axis) and Treg cells (y-axis). Differential interactions between Treg and Tcon cells are labeled red (FDR 1%). **b** Hi-C browser shot of a region showing Treg specific interactions over a ~800 kb locus containing a cluster of Treg upregulated genes (*Plxnc1*, *Socs2*, *Ube2n*, *Nudt4*). Below the Hi-C data is a genome browser track of ChIP-seq for CTCF (green), Smc1a (purple), H3K27ac (blue), and Foxp3 (orange, showing ChIP-seq and Cut&Run) in Treg and Tcon cells. **c** Plot of transcript abundance (x-axis) versus fold change (y-axis) of gene expression between CD4⁺ Tcon cells and Treg cells. Differentially expressed genes are shown in red (FDR 1%, minimum 2-fold change). **d** Bar plots showing the percentage of genes by each class that overlap either Treg (left) or Tcon (right) specific chromatin interactions. The classes of genes considered are those that are Treg specific (blue), Tcon specific (yellow), or non-differentially expressed genes (gray). (*p*-values are from two-sided Fisher's exact test). **e** Histogram of the number of

differential interacting partners for each differential interaction anchor. Most loci are involved in differential interactions with one partner anchor, but a minority of loci form multiple differential long-range contacts. **f** Fraction of anchors of differential chromatin interactions that overlap differentially expressed genes for loci involved in one or more differential interactions. Loci involved with multiple differential long-range interactions are more likely to contact differentially expressed genes. **g** Heat map of genes overlapping anchors involved in >4 differential chromatin interactions (log2 fold-change). **h** Example of multi-way long-range differential chromatin interaction. Treg chromatin interactions are shown in the upper right portion of the heat map, and Tcon chromatin interactions are shown in the lower left portion of the heat map. **i** Enrichment of ChIP-seq peaks over differential chromatin interactions. ChIP-seq data is shown either from CD4⁺ Tcon cells (pink box) or Treg cells (blue box). Enrichment is shown for Tcon specific interactions (yellow bars) or Treg specific interactions (blue bars).

## Foxp3 is critical for the establishment of Treg-specific 3D chromatin structure

Foxp3 is a pivotal regulator of Treg differentiation and function. Given that Foxp3 bound peaks are highly enriched in the anchor regions of Treg-specific chromatin interactions, we examined Foxp3's role in the formation and maintenance of Treg-specific DNA loops. To this end, we used a GFP knock-in Foxp3 knockout mouse strain (Foxp3^GFP-KIKO)[33], in which GFP expression replaces Foxp3 and can be used as a fluorescent marker to identify Treg "wannabe" cells (Fig. 3a). As reported previously, Treg "wannabe" cells receive similar TCR signals and developmental cues like WT Tregs, express the majority of Treg signature genes, and might represent Treg precursors, as reflected by mRNA transcriptional activity at the *Foxp3* locus but lack Foxp3 protein expression and suppressive function[29,33]. Hi-C experiments were performed with GFP⁺Foxp3⁻ Treg- "wannabe" cells from healthy heterozygous female Foxp3^GFP-KIKO/WT mice and control GFP⁺Foxp3⁺ WT Treg cells from Foxp3^GFP/WT mice. We focused on chromatin contacts that were Treg specific in our comparison of Treg and Tcon chromatin interactions. Comparison between the chromatin contacts in GFP⁺Foxp3⁺ WT Treg cells and GFP⁺Foxp3⁻ Treg "wannabe" cells showed that a total of 94 out of 1959 Treg-specific interactions were significantly reduced in the GFP⁺ Foxp3-KIKO cells (Fig. 3b, FDR = 10%). While this represents a minority of the Treg-specific contacts (4.8% or 94/1959), the decrease in Treg-specific DNA contacts in Foxp3 KIKO cells was also illustrated by the overall distributions of KIKO vs. WT contact frequency fold-changes (Fig. 3c). Specifically, 76.3% (1495/1959) of Treg-specific contacts showed a decrease (fold change <0) in chromatin interaction frequency in GFP⁺ Foxp3-KIKO cells relative to control GFP⁺Foxp3⁺ WT Treg cells. Similarly, Tcon-specific chromatin interactions were globally upregulated in Foxp3 KIKO cells compared to WT Tregs (Fig. 3c). We performed RNA-seq to identify genes that change expression between Foxp3-KIKO and WT Tregs. We identified 566 differentially expressed genes between Foxp3-KIKO and WT Tregs (FDR < 5%, fold-change >2) (Supplementary Data 3). Of these differentially expressed genes, these were strongly enriched for overlapping with sites of differential chromatin interactions between Treg and Tcon cells or between Foxp3-KIKO and WT Tregs (Fig. 3d), suggesting that the change in chromatin interactions between Foxp3-KIKO and WT Tregs likely contributes to altered patterns of expression in a Foxp3 dependent manner.

The reduction of chromatin interactions in GFP⁺Foxp3⁻ Treg- "wannabe" cells could be seen at the locus of the *Ikzf2* gene, one of the Treg signature genes that was significantly downregulated in Foxp3-KIKO Tregs. In Tcon cells, there were no detectable chromatin interactions around *Ikzf2*, while strong DNA interacting loops emerged in WT Treg cells. In the Treg- "wannabe" cells, the chromatin interactions were significantly weakened at the *Ikzf2* locus (Fig. 3e). ChIP-seq and Cut&Run experiments showed the DNA looping anchors around *Ikzf2* were enriched with Foxp3 bound peaks along with CTCF and cohesin

component Smc1a bound peaks (Fig. 3f). The fact that some but not all Treg specific chromatin interactions lost in GFP⁺Foxp3⁻ Treg- "wannabe" cells led us to investigate whether contacts lost in GFP⁺Foxp3⁻ Treg- "wannabe" cells may be more dependent on Foxp3. Indeed, chromatin contacts that were lost in the Treg- "wannabe" cells compared to controls were more likely to contain Foxp3 binding peaks (Fig. 3g). In addition to Foxp3 binding sites, we also examined the enrichment of TF motifs in Treg DNase I Hypersensitive Sites (DHS) and identified motifs with differential TF motif enrichment in chromatin contacts lost versus retained in GFP⁺Foxp3⁻ Treg- "wannabe" cells versus control cells (Fig. 3h). Specifically, we saw modest enrichments of CTCF and ETS-family TF motifs in DHS sites at chromatin contacts retained in GFP⁺Foxp3⁻ Treg- "wannabe" cells, whereas we saw enrichment of AP-1 and Forkhead motifs in the chromatin contacts lost in the GFP⁺Foxp3⁻ Treg- "wannabe" cells. This suggests that these TFs, in conjunction with Foxp3, may be critical for the establishment of the mature Treg 3D chromatin landscape. Taken together, these data suggest that the establishment of Treg-specific chromatin interactions is dependent on Foxp3 expression.

## Mutations in the Foxp3 domain-swapped dimerization interface lead to inadvertent immune system activation in mice

Although our data clearly indicated that Foxp3 is indispensable for establishing Treg-specific 3D chromatin structure, it is not clear whether Foxp3 is directly involved in DNA looping like CTCF/cohesin, or whether it acts indirectly as a factor that facilitates the binding of architectural proteins including CTCF and cohesin to form Treg-specific chromatin interactions. It was reported that Foxp3 is able to form a domain-swap (DS) dimer through its forkhead domain to bring two distal DNA elements together[31], suggesting Foxp3 has the potential to facilitate DNA looping directly (Fig. 4a). Of note, three amino acid mutations (W348Q, M370T and A372P) in the DS interface that disrupt DS dimer formation diminish Treg suppressive activity without compromising Foxp3 DNA binding in vitro, indicative of a direct role of Foxp3 in regulating Treg function by modulating Treg cell 3-D chromatin structure[31]. To test this possibility, we generated a Foxp3 DSM mouse strain by using CRISPR technology to introduce the above mutations (W348Q, M370T and A372P) into the Foxp3 coding region in the Foxp3-IRES-Thy1.1 reporter mouse (Fig. 4b). After verifying the mutations by DNA sequencing, we analyzed Foxp3 DSM mice and WT littermate controls to determine whether disabling Foxp3's domain swapped dimerization affects Treg cell development and function. The Foxp3 DSM mice appeared to be normal up to 2 months of age when they started to develop a moderate lymphoproliferative disease with loss of body weight and increased cellularity in the spleen (Fig. 4c, d). The frequency of Treg cells was higher while Foxp3 protein level was significantly reduced in Foxp3 DSM mice compared to WT controls (Fig. 4e). CD4⁺ conventional T cells and CD8⁺ T cells were also more activated with the expansion of the CD44⁺CD62L^low population.

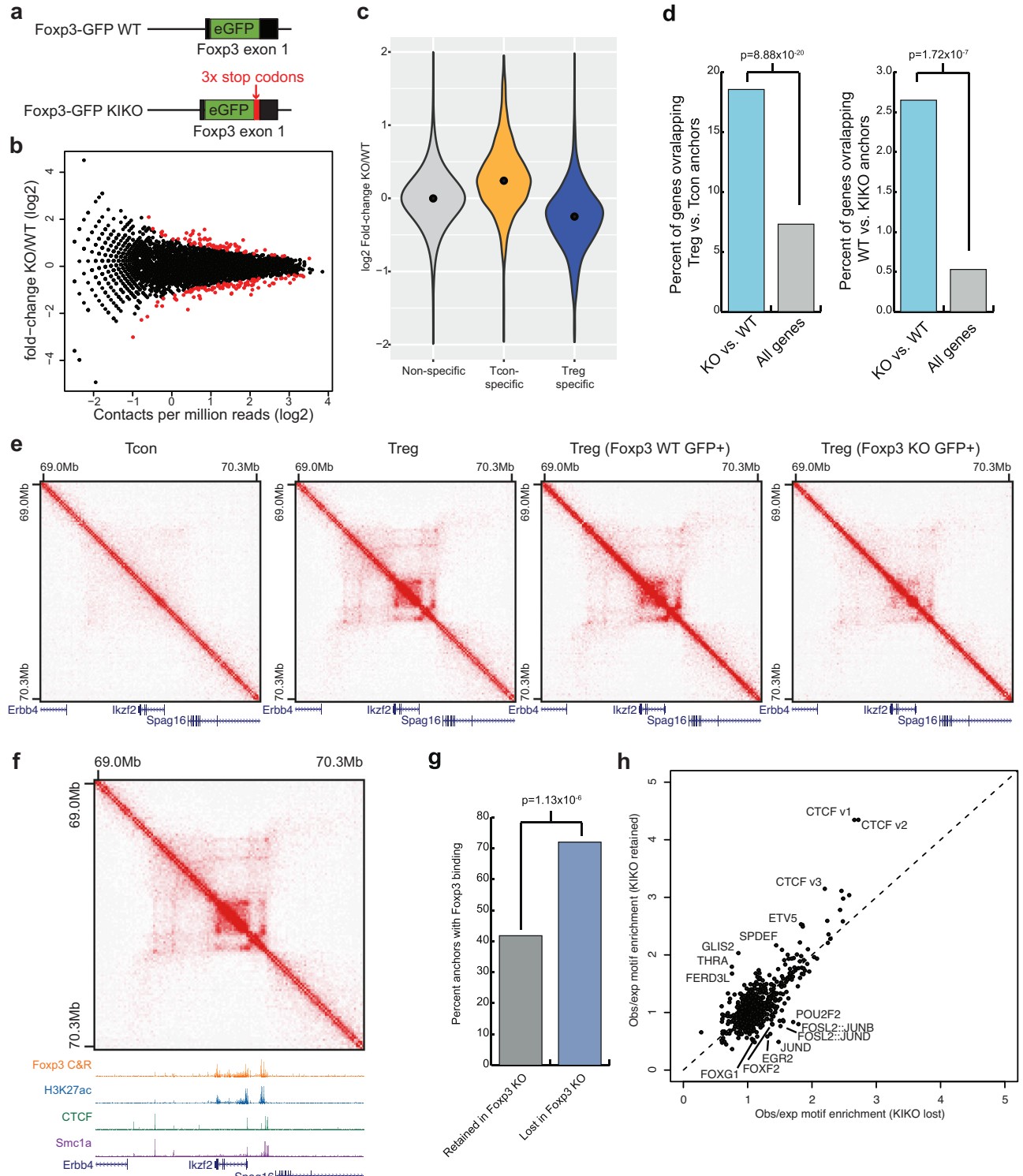

(Fig. 4f). Consistently, IFNγ production increased significantly in splenic CD4+ T cells from Foxp3 DSM mouse (Fig. 4g). Furthermore, serum concentrations of IgG1 and IgM were significantly higher in Foxp3 DSM mice compared to WT controls (Fig. 4h). Histopathology analysis of 6- to 9-month-old Foxp3 DSM mice revealed widespread immune cell infiltration in the lung, liver, small intestine, and salivary gland tissues (Fig. 4i). Taken together, these data suggested that the inability of Foxp3 to dimerize in "trans" leads to a moderate defect in Treg cell's immune suppressive function which results in excessive immune system activation.

## The formation of Treg-specific chromatin interactions is independent of Foxp3 domain-swap dimerization

Next, we sought to assess whether the impairment of Foxp3 DSM Treg function was because DS mutation disrupted the formation of Treg 3D genome structure, and subsequently affected Treg signature gene expression. To this end, we isolated Treg cells from 6-week-old asymptomatic Foxp3 DSM and control mice and performed in-situ Hi-C experiments to map their 3-D genome structure. Foxp3 DSM and WT Treg cells showed similar chromatin interaction patterns as a whole and within Treg-specific interactions (Fig. 5a). In fact, none of the 1959

**Fig. 3 | Foxp3 is required for the complete establishment of Treg specific 3D chromatin structure. a** Schematic of the Foxp3-GFP knock-in and the Foxp3-GFP (WT) and Foxp3 knock-in/knock-out (KIKO) mice. eGFP is knocked into the *Foxp3* locus in both the Foxp3-GFP and KIKO mice. In KIKO mouse, the eGFP is followed by 3x stop codons to prevent Foxp3 protein translation. **b** Chromatin interaction frequency between WT and KIKO Treg cells at $N = 1569$ Treg specific chromatin interactions. Treg specific chromatin interactions were identified by comparison of splenic Treg and Tcon cells as shown in Fig. 2. Differential chromatin interactions ($N = 125/1569$) are labeled as points in red (FDR = 10%). **c** Violin plot of the log2 fold-change in interaction frequency between Foxp3 WT and KIKO Treg cells at Treg specific chromatin interactions (blue) or Tcon specific chromatin interactions (yellow). The median log2 fold change of Treg specific chromatin interactions is −0.24 (black dot) in KIKO Tregs vs. WT, consistent with a loss of chromatin interactions in KIKO Tregs at Treg specific chromatin interactions. **d** Bar plots showing

the percentage of differentially expressed genes (blue) compared with all genes (gray) that overlap Treg vs. Tcon specific chromatin interactions (left) or that overlap differential interactions between Foxp3 WT and KIKO Treg cells (right) (*p*-value is from two-sided Fisher's exact test). **e** Hi-C interaction maps at the *Ikzf2/Helios* gene in Tcon, Treg, Foxp3 WT Treg, and Foxp3 KIKO Tregs (left to right). **f** Chromatin interactions in Treg cells at the *Ikzf2/Helios* locus and the associated ChIP-seq or CUT&RUN experiments over the locus. **g** Enrichment of Foxp3 CUT&RUN peaks over Treg specific chromatin interactions that are either retained in KIKO cells (gray) or lost in KIKO cells (blue) showing that the regions that lose chromatin interactions in Foxp3 KIKO cells are more likely to have Foxp3 binding sites (*p*-value is from two-sided Fisher's exact test). **h** Observed vs. expected motif frequency in DNase I Hypersensitive sites overlapping chromatin interactions lost in Foxp3 KIKO cells (x-axis) vs. chromatin interactions retained in Foxp3 KIKO cells (y-axis).

Treg specific chromatin interactions is significantly different in the DSM Tregs compared to the WT Tregs (FDR < 10%) (Fig. 5b). Furthermore, we did not observe a global loss of chromatin contacts at Treg-specific interactions in the DSM Treg cells as we do in the GFP⁺ Foxp3 KIKO cells (Fig. 5c). Finally, the lack of differences in 3D structure between DSM and WT Tregs was not related to statistical power or data quality, as the sequencing depth and Hi-C library quality was comparable between Tcon, Treg, KIKO, and DSM experiments (Supplementary Data 1). Taken together, these results show that domain-swapped dimerization of Foxp3 is not required for establishing Treg-specific 3-D genome structure.

To further dissect the molecular mechanism underlying DSM Treg's impaired function, we performed RNA-seq experiments with DSM and WT Treg cells. There were 26 up-regulated genes and 121 down-regulated genes in DSM Tregs compared to WT Tregs (Fig. 5d, FDR 5%, greater than 2-fold change, Supplementary Data 4). Interestingly, the differentially expressed genes in the DSM Treg cells showed expression patterns resembling Tcon cells, rather than WT Treg cells (Fig. 5e), suggesting that the DS mutation disrupts Foxp3's transcriptional regulation function. The expression of several Treg signature genes, *Lrrc32* (encodes the GARP protein), *Tigit*, and *Ctla4*, were compromised in DSM Tregs, contributing to their impaired immune suppressive function.

To further explore the mechanism underlying dysregulated gene expression in DSM Tregs despite the lack of changes in 3D genome structure, we examined whether the DSM mutation contributed to differences in Foxp3 binding using CUT&RUN. We identified 36,727 Foxp3 binding peaks in WT and DSM Tregs by CUT&RUN. Comparing peak strength between WT and DSM-mutant Tregs, we identified 295 peaks that showed differential Foxp3 binding (Fig. 5f, FDR 5%, Supplementary Data 5). We then analyzed differential Foxp3 binding peaks based on their distance to differentially expressed genes (DEGs) in WT and DSM Tregs. The differential DSM Foxp3 binding peaks were significantly closer to DEGs compared to non-affect Foxp3 binding peak controls (Fig. 5g). For example, although chromatin interactions around *Lrrc32* were similar, the main Foxp3-bound peak was weaker in DSM Tregs compared to WT controls (Fig. 5h, i). It is likely that defective Foxp3 binding resulted in the reduced expression level of Lrcc32 in DSM Tregs (Fig. 5j). Taken together, these results suggest that the domain-swapped mutation negatively affects Foxp3 binding to the loci of a subset of genes, leading to compromised gene expression and defective Treg function.

## Foxp3-associated chromatin interactions are functionally required for Treg cell function

To further understand how Foxp3 is involved in the establishment of Treg-specific chromatin interactions, we performed proximity ligation-assisted ChIP-seq (PLAC-seq)[36] with an antibody against Foxp3 in Treg cells. PLAC-seq is an assay that combines Hi-C experiments with

ChIP, such that chromatin interactions associated with a factor of interest can be studied in higher resolution. This enabled us to analyze Foxp3-associated chromatin interactions with higher sensitivity and efficiency. Using Foxp3 PLAC-seq data, we identified 1165 focal loops at 10 kB resolution (Supplementary Data 6). These focal loops were highly overlapped with Foxp3 binding peaks in Treg cells (Fig. 6a). In total, we identified 2169 Foxp3-associated loop anchors, of which the majority contained a Foxp3 binding site or had a Foxp3 binding site within 10 kB (Fig. 6b). Of note, 9.01% of the total Foxp3 binding sites are associated with chromatin interactions identified by Foxp3 PLAC-seq (Fig. 6c). We next classified Foxp3-associated DNA loops based on the locations of the loop anchors relative to genes. More than two-thirds of the loops are gene-gene (22.49% vs. expected 4.35%), gene-enhancer (27.12% vs expected 2.61%), enhancer-enhancer (17.17% vs. expected 1.14%) interactions, implicating their involvement in the regulation of gene expression (Fig. 6d).

To examine how Foxp3-associated chromatin interactions influence gene expression, we compared the expression of genes whose transcription starting site overlaps with Foxp3 PLAC-seq interactions between CD4+ conventional T cells and Tregs. A number of Treg signature genes, including *Ikzf2*, *Icos*, *Entpd1*, and *Il7r*, emerged as associated with Foxp3 PLAC-seq interactions (Fig. 6e). We next directly tested whether Foxp3-associated chromatin loops facilitate gene expression at the *Ikzf2* locus. Using a pair-guide RNA mediated CRISPR/Cas9 approach[37], we systemically deleted 9 Foxp3 binding sites (named P1 to P9) located close to the *Ikzf2* gene, and measured Helios protein (encoded by the *Ikzf2* gene) expression in Treg cells compared to cells transduced with non-targeting sgRNAs (Fig. 6f and Supplementary Fig. 2). When the Foxp3 binding site at the P1 or P6 regions were deleted, Helios expression decreased significantly (Fig. 6g). The P6 region is located in the *Ikzf2* promoter, therefore its deletion served as a positive control. We next deleted the P1 or P6 region in mature splenic Treg cells in vitro, which were then transferred into Rag1 knockout recipient mice to track deletion's impact on Tregs in vivo. Indeed, Helios expression was reduced in Tregs isolated from P1 deleted Treg cells (Fig. 6h), suggesting that the P1 region is a Foxp3 bound enhancer. Interestingly, Hi-C analysis showed that P1 deletion did not significantly weaken Foxp3 PLAC-seq peak at the *Ikzf2* locus (Fig. 6i). This data suggests that Foxp3 may not be required for the maintenance of Treg-specific chromatin interactions, despite that Foxp3 binding is essential for Ikzf2 expression.

To further characterize Foxp3's role in the maintenance of Treg-specific chromatin interactions in mature Treg cells, we used CRISPR/Cas9 to knockout Foxp3 in splenic Treg cells in vitro and profiled their 3D genome structure by in situ Hi-C (Supplementary Fig. 3a, b). Unlike GFP⁺Foxp3⁻ KIKO cells, the deletion of Foxp3 by sgRNA in mature Treg cells did not change their chromatin interaction patterns as a whole or within Treg-specific interactions (Supplementary Fig. 3c). There was no global loss of chromatin contacts at Treg-specific interactions in the

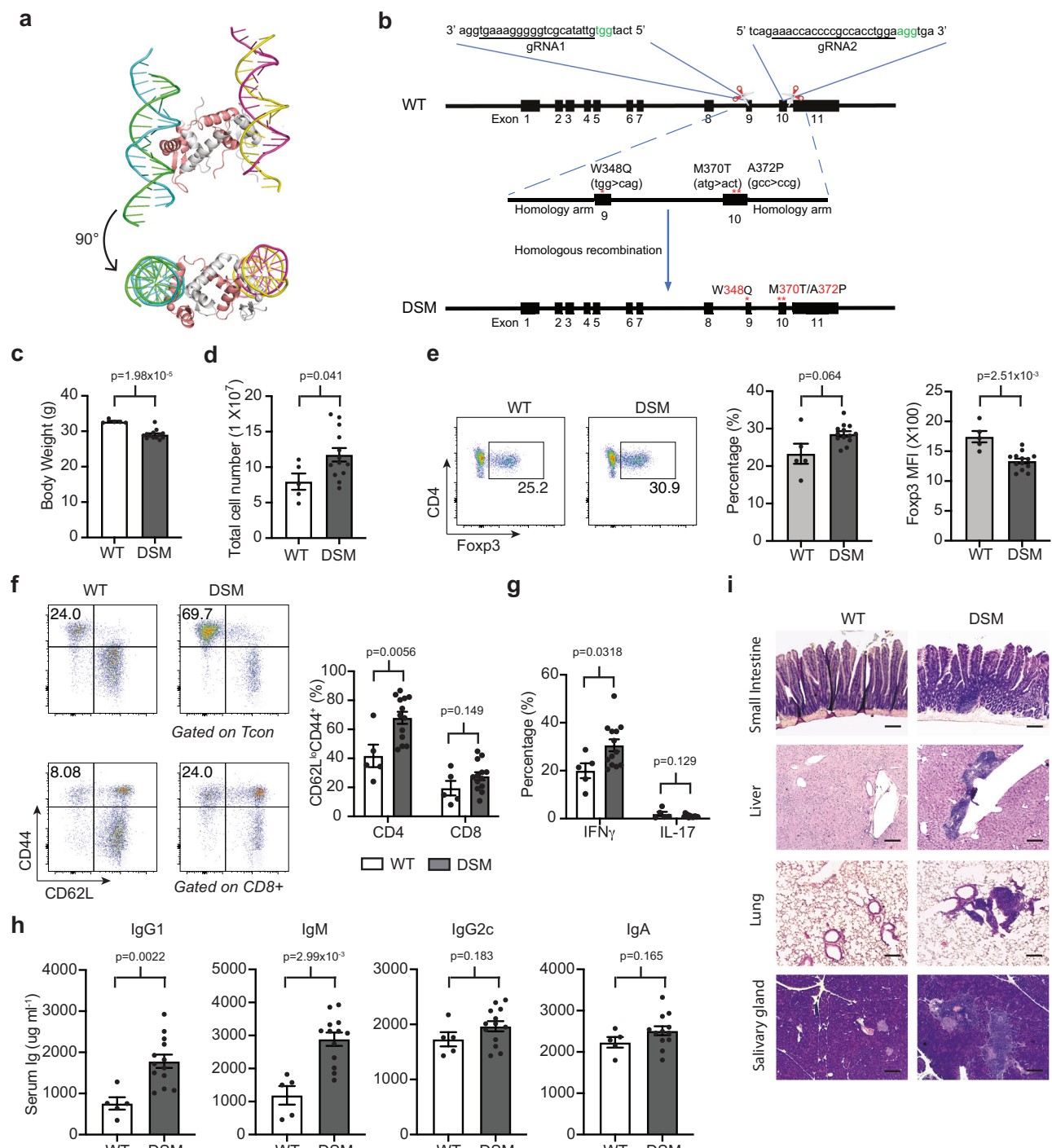

**Fig. 4 | Foxp3 domain swap mutant mice develop an autoinflammatory phenotype. a** Schematic of Foxp3 domain swap dimer. Two Foxp3 proteins interact as a dimer (white and pink chains) binding to two DNA strands in a "trans" configuration. Structure from PDB accession 3QRF. **b** Strategy for generation of Foxp3 domain swap mutant mice by CRISPR/Cas9. Two sgRNAs were used to cut off the endogenous genomic fragment, and a synthesized single-strand DNA with domain swap mutation was used to guide CRISPR/Cas9 mediated homology-directed repair. **c**–**i** 6- to 9-month-old Foxp3 DSM ($n=13$) and WT ($n=5$) littermate control mice were analyzed for (**c**) Body weight, (**d**) spleen cellularity, (**e**) Frequency of

Foxp3⁺ Treg cells in total CD4⁺ T cells and Foxp3 protein level. **f** frequency of CD44⁺CD62L^low cells in CD4⁺ Tcon and CD8⁺ T cells, (**g**) frequency of IFNγ⁺ and IL-17⁺ cells in CD4⁺ Tcon cells. **h** ELISA quantification of the concentrations of IgG1, IgM, IgG2c, and IgA in serum. **i** Representative hematoxylin and eosin staining of small intestine, liver, kidney, and lung sections from WT and Foxp3 DSM mice, Scale bars, 200 μm. *p*-values were calculated by an unpaired, two-sided Student's *t* test (**c**–**h**). Data are represented as mean ± s.e.m. Source data are provided as a Source Data file.

sgFoxp3 Treg cells as in the GFP⁺ Foxp3⁻ KIKO cells (Supplementary Fig. 3d). Therefore, Foxp3 is essential for establishing Treg-specific chromatin interactions but dispensable for the maintenance of these interactions in mature Treg cells.

Our results comparing Foxp3-KIKO vs. WT Treg cells suggest that Foxp3 is critical for the establishment of Treg-specific chromatin interactions. In addition, chromatin interactions that are lost in Foxp3-KIKO cells show different motif enrichment patterns compared with

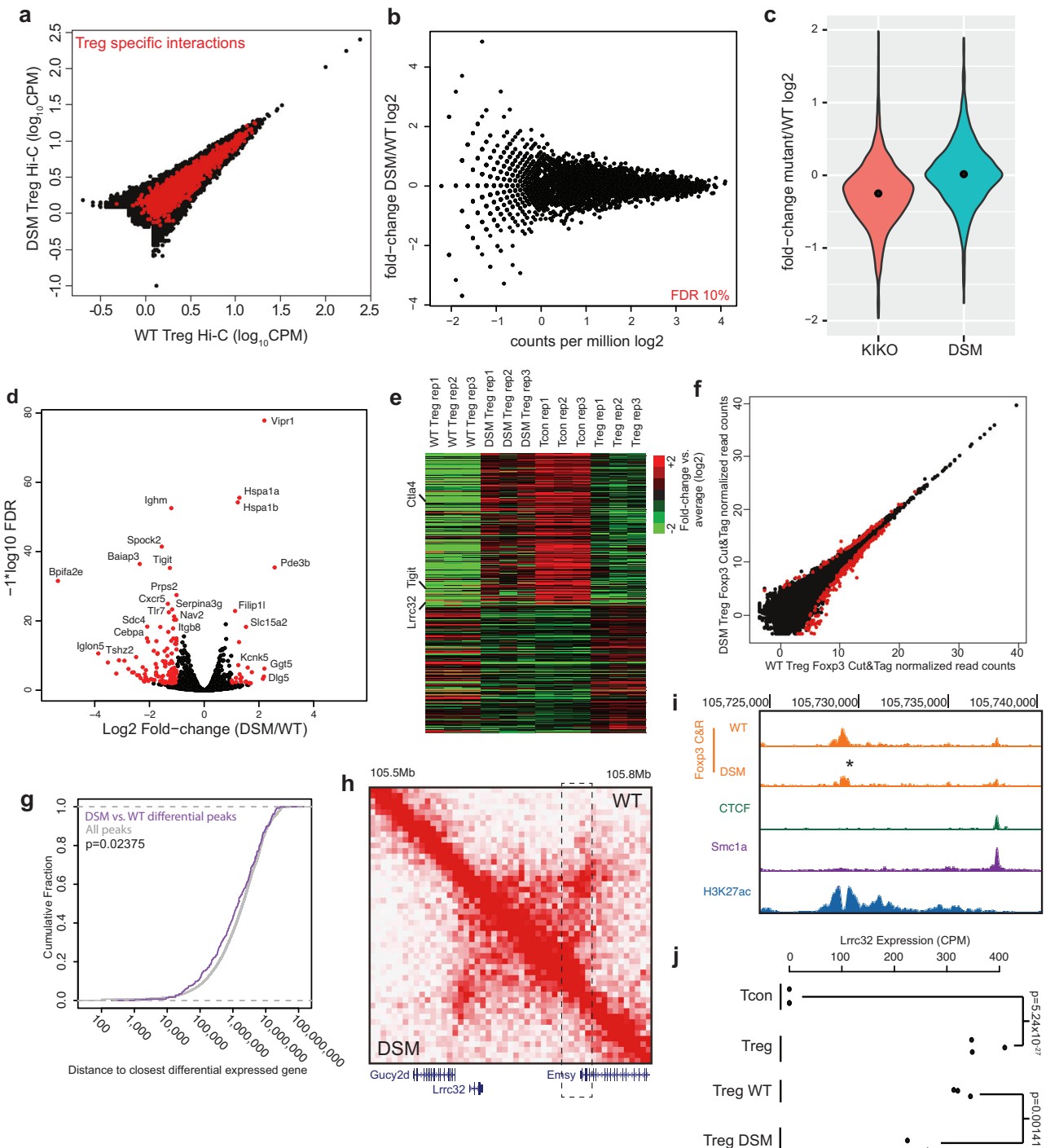

**Fig. 5 | Foxp3 domain swap mutant and its impact on 3D genome structure and gene expression. a** Hi-C chromatin interactions in wild-type (WT) Treg cells (x-axis) and Foxp3 domain swap mutant (DSM) Treg cells (y-axis). Treg specific interactions between Treg and Tcon cells are in red. **b** Total chromatin interaction frequency (x-axis) versus fold change (log2) in chromatin interaction frequency (y-axis) between DSM and WT Treg cells. Points in red are significant differential interactions (FDR 10%). **c** Violin plot of the fold change (log2) in interaction frequency between Foxp3 WT and Foxp3 KIKO Treg cells (red) or between Foxp3 WT and Foxp3 DSM Treg cells (green) over Treg specific chromatin interactions. The black dots show the median fold-change. **d** Volcano plot of RNA-seq between DSM and WT Treg cells. Points in red are significantly differentially expressed genes between DSM and WT Tregs (FDR 1%, 2 fold minimum change). **e** Heat map of expression in CD4+ Tcon cells and different WT or DSM mutant Treg cells over genes differentially expressed between CD4+ Tcon cells and Treg cells. **f** Foxp3 CUT&RUN peak enrichment in

Foxp3 WT Treg (x-axis) and Foxp3 DSM Treg cells (y-axis). Points in red ($N = 295$) are differential Foxp3 peaks (FDR 5%). **g** Cumulative density plot of the distance between differential Foxp3 peaks and differentially expressed genes (purple) compared to the distance between all Foxp3 peaks and differentially expressed genes (gray) showing that Foxp3 peaks affected by the DSM mutation are closer to DSM dysregulated genes (p-value is calculated by two-sided Wilcoxan Rank Sum test). **h** Chromatin interactions near the *Lrrc32* gene in Foxp3 WT Treg cells (upper right) versus Foxp3 DSM Treg cells (lower left). The dashed line box shows a distal enhancer with a Foxp3 peak with reduced binding in DSM Treg cells. **i** Enhancer in the dashed box from panel H showing the reduction in Foxp3 binding in DSM Treg cells by Foxp3 CUT&RUN. **j** Expression of *Lrrc32* in Tcon, Treg, Foxp3 WT Treg, and Foxp3 DSM Treg cells showing reduced *Lrrc32* expression in DSM Treg cells (p-value is from Benjamini corrected quasi-likelihood F-test in edgeR).

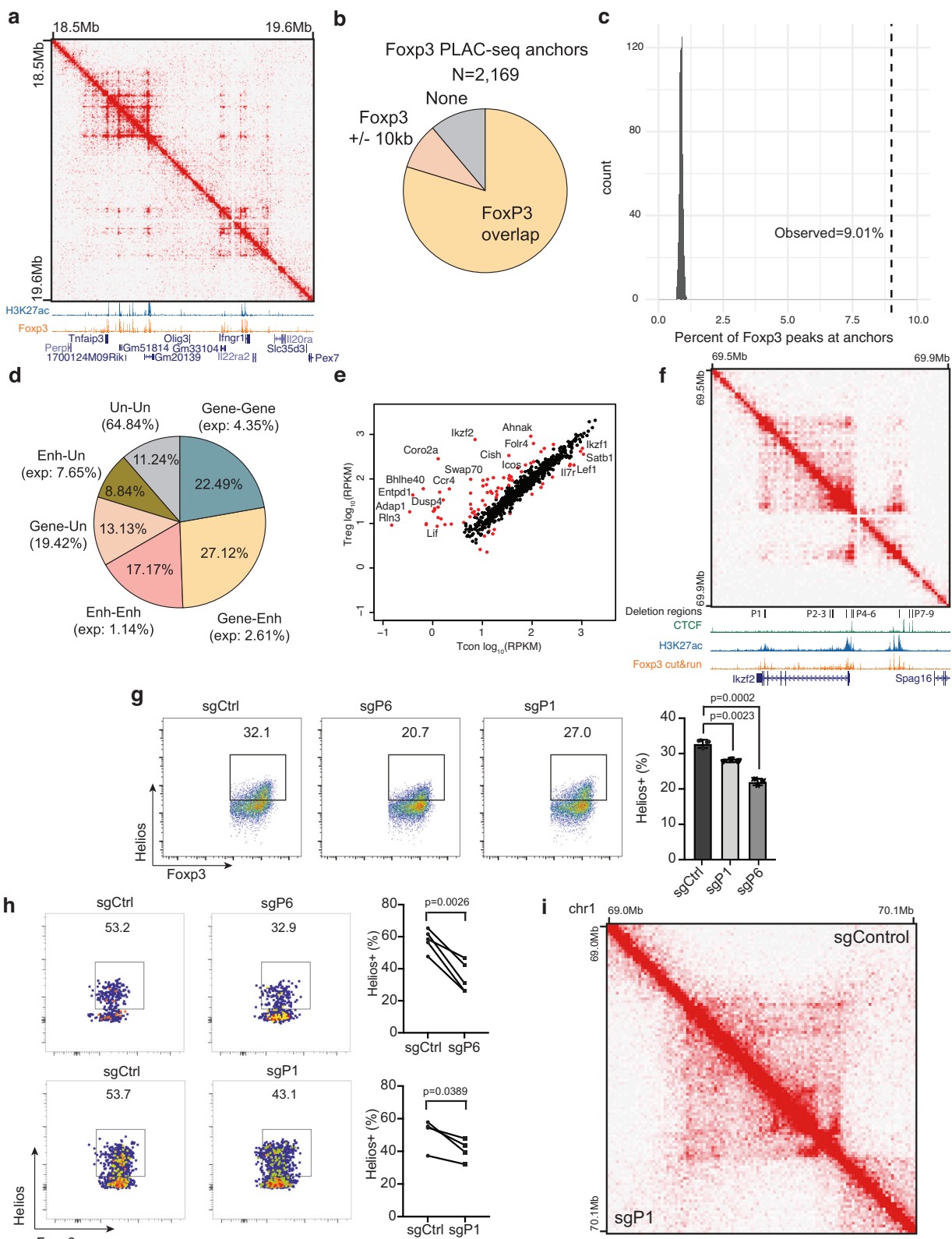

chromatin interactions that are retained (Fig. 3h). This raises the possibility that Foxp3 may control a subset of chromatin interactions, and therefore genes, during Treg lineage commitment. To further investigate Foxp3's role in Treg gene expression, we compared Treg-specific genes, which are differentially expressed between Treg and Tcon cells, with genes that are dysregulated in either the KIKO or DSM Treg cells. This divided the Treg-specific genes into two sets, "Foxp3 dependent"

(changing in KIKO or DSM or both) and Foxp3 independent (not changing in KIKO or DSM) (Fig. 7a, Supplementary Data 6). Gene Ontology (GO) analysis of these two sets of genes revealed distinct GO terms. For the Foxp3-dependent genes, the top terms were related to immune function, cytokines, and cell adhesion (Fig. 7b). For the Foxp3 independent genes, the top hits were related to metabolism and cell cycle (Fig. 7c). This was not caused by a specific *p*-value threshold,

**Fig. 6 | Foxp3-associated chromatin interactions link distal enhancers to Treg signature genes. a** Chromatin contact map of Foxp3 PLAC-seq data near the Treg specific genes *Tnfaip3* and *Ifngr1*. Shown below are genome browser tracks of H3K27ac ChIP-seq (blue) and Foxp3 CUT&RUN (orange). **b** Pie chart showing the fraction of Foxp3 PLAC-seq anchors containing a Foxp3 binding site (yellow), within 10 kb of a Foxp3 binding site (+/− 1 bin, tan), or without a Foxp3 binding site (gray). **c** Plot showing the fraction of Foxp3 binding sites associated with PLAC-seq interactions. The observed overlap (9.01%) is shown as a dashed line compared to random permuted overlaps (1000 iterations, gray). **d** Pie chart showing the fraction of Foxp3 PLAC-seq peaks that link genes (Gene), enhancers (Enh), or unannotated regions (Un). Enhancers are defined as distal H3K27ac peaks. Unannotated regions are sites that do not overlap gene transcription start sites or distal H3K27ac peaks. **e** Gene expression in CD4+ Tcon (x-axis) and Treg cells (y-axis) of genes whose TSS overlaps with Foxp3 PLAC-seq interactions. Points in red are differentially

expressed genes between Treg and Tcon cells (FDR1%, 2-fold minimum change). **f** Foxp3 PLAC-seq data near the *Ikzf2*/Helios locus. Shown below are ChIP-seq and CUT&RUN tracks and the locations of Foxp3 binding sites targeted for CRISPR deletion (P1-P9). **g** The effect of CRISPR deletion of Foxp3 binding sites P1 or P6 on Helios expression in Treg cells in vitro. *p*-values were calculated by an unpaired, two-sided Student's *t* test. **h** In vivo validation of the effect of P1 or P6 deletion on Helios expression. Treg cells transduced with gRNAs targeting P1 (*n* = 4) or P6 (*n* = 5) were co-transferred with Tregs with control gRNAs into the same Rag1-/- mice, and Helios expression was analyzed 2 weeks post transfer. *p*-values were calculated by a paired, two-sided Student's *t* test. Data are represented as mean ± s.e.m. Representative data of two independent experiments (**g**, **h**) are shown. Source data are provided as a Source Data file. **i** Hi-C interaction map at the Ikzf2/Helios gene in Treg cells transduced with control gRNAs (upper right) or gRNAs targeting P1 (lower left).

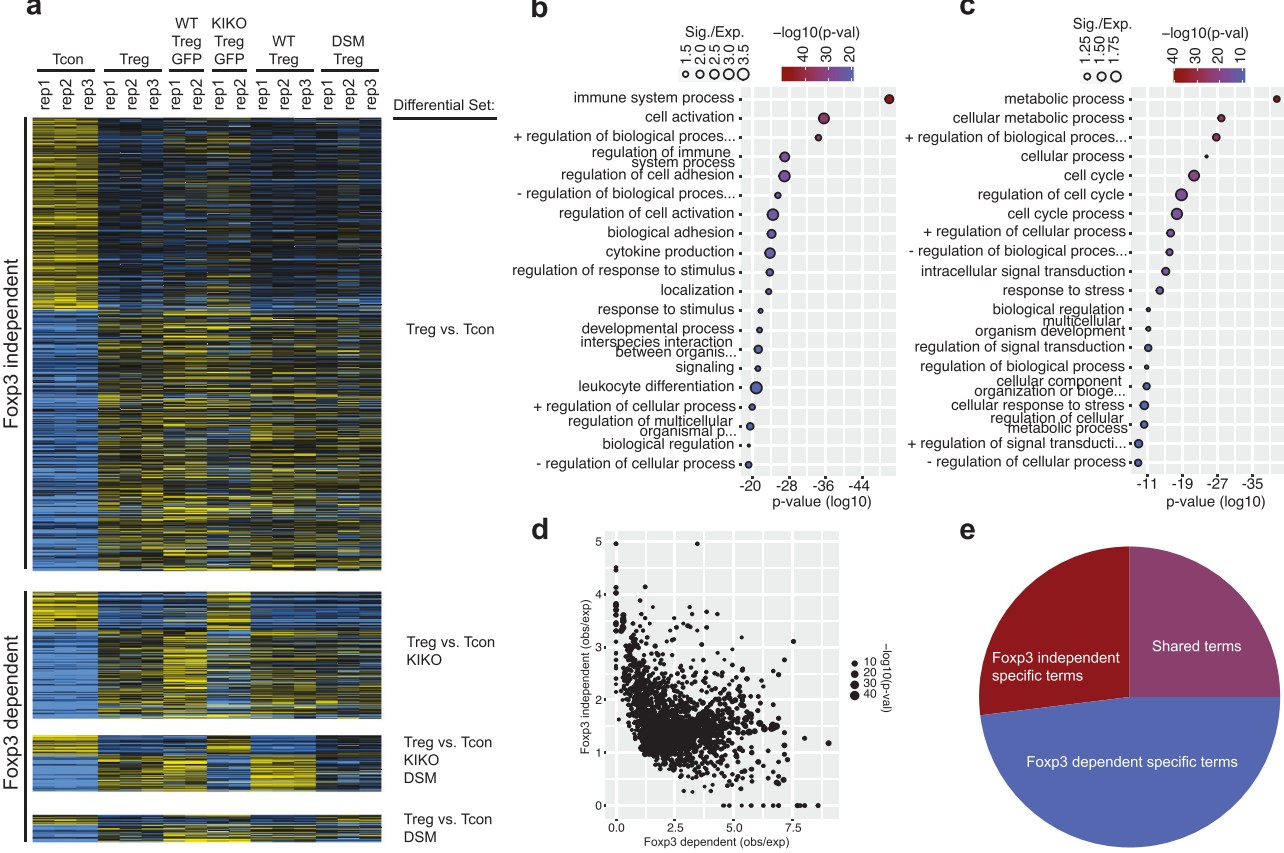

**Fig. 7 | Foxp3 dependence defines differential programs of Treg lineage commitment. a** Gene expression of genes differentially expressed between Tcon and Treg cells across Foxp3 WT and mutant cells. Genes are separated into those that show significant differences in the Foxp3 KIKO, DSM, or both mutants ("Foxp3 dependent") and those that show differential regulation in Tcon vs. Treg cells but do not show differences in the Foxp3 mutant cells ("Foxp3 independent"). **b** Gene ontology analysis of the "Foxp3 dependent" genes shows terms related to immune differentiation and function (*p*-value is from classicFisher function in topGO).

**c** Gene ontology analysis of "Foxp3 independent" genes shows terms related to metabolic regulation and the cell cycle (*p*-value is from classicFisher function in topGO). **d** Comparison of the observed/expected Gene ontology term frequency for Foxp3 dependent (x-axis) versus independent (y-axis) genes shows an inverse correlation of enrichment of ontology terms (*p*-value is from classicFisher function in topGO). **e** Pie chart of gene ontology terms that are significant for either "Foxp3 dependent" or "Foxp3 independent" genes that are significant in both sets (purple) or Foxp3 independent alone (red) or Foxp3 dependent alone (blue).

because by comparing the GO terms obs/exp gene sets, there were a large number of GO terms that were specific to the Foxp3 dependent or independent set (Fig. 7d). Furthermore, about 3/4 of all the GO terms that came up as significant in either the Foxp3 dependent or independent sets are specific to each set (Fig. 7e). These analyses suggested that Foxp3 regulates genes related to the immune function of Treg cells, while other factors regulate cell cycle and cell metabolic processes during Treg lineage commitment.

## Discussion

In this study, we examined the role of Treg lineage-specific TF Foxp3 in 3D genome organization during the late stage of T cell development. We compared the 3D chromatin structures of Treg cells and their precursors and revealed that the 3D chromatin architecture of Treg cells was gradually established during Treg lineage specification, and that changes in chromatin interactions align with the trajectory of Treg development. By comparing Treg cells and their closely related

conventional T cells, we identified chromatin structures unique to Treg cells. Overall, the Treg's chromatin structure was highly similar to that of conventional T cells, 0.29% (3932/1377570) chromatin interactions were significantly different between Treg and Tcon cells. This small number of differential chromatin interactions are in line with less than 1% of differential enhancers, and less than 5% of differentially expressed genes between Treg and conventional T cells. Strikingly, these Treg-specific chromatin interactions were frequently associated with the loci of Treg signature genes.

Our study investigated Foxp3's contribution to the establishment and maintenance of Treg chromatin structure. Notably, Foxp3 was significantly enriched at the anchor regions of Treg-specific chromatin loops, even ranking higher than CTCF/cohesin. By taking advantage of Treg-"wannabe" cells from Foxp3[GFP-KIKO] mice, our data revealed that Foxp3 was essential for the establishment of Treg-specific chromatin structure, which agreed with a recently published Foxp3 HiChIP study[30]. It was proposed that the domain-swapped dimer of Foxp3 facilitates its potential role as a DNA loop anchor to establish Treg-specific chromatin interactions[31]. We further examined whether Foxp3 can function as a DNA loop anchor by comparing of WT and DSM Treg cells. Although the Foxp3 DSM Treg cells were dysfunctional and the Foxp3 DSM mice developed lymphoproliferative disease, the 3D chromatin architecture of the Treg cells was not affected by mutations disrupting Foxp3 domain-swapped dimerization. Instead, the defective function of DSM Tregs is likely caused by impaired Foxp3 binding to key Treg gene loci. Furthermore, a recent study showed that the Foxp3 domain-swapped dimer is dysfunctional, while a head-to-head dimer represents the physiological form of Foxp3[38]. Altogether, these data suggested that domain-swapped dimerization of Foxp3 is not required for the organization of Treg's 3D chromatin structure.

The Foxp3 PLAC-seq and Foxp3 ChIP-seq/CUT&RUN data also revealed that Foxp3-associated chromatin interactions were associated with the loci of Treg-signature genes, including *Ikzf2*, *Icos*, *Entpd1*, and *Il7r*. Although the deletion of Foxp3 binding sites in the *Ikzf2* locus decreased its expression, it did not disrupt the chromatin interactions at the corresponding regions. Furthermore, the ablation of Foxp3 in mature Treg cells did not alter their 3D genome structure. This data indicates that although Foxp3 plays an essential role in the formation of Treg-specific chromatin interactions, it has only marginal effects on the maintenance of their 3D chromatin structure. Furthermore, by comparing WT Treg, "wannabe" Treg, and DSM Treg gene expression, we found that Treg-characteristic genes can be separated into Foxp3-dependent and Foxp3-independent groups, with Foxp3-dependent genes enriched in Treg immune suppressive functions, while Foxp3-independent genes enriched in metabolic and cell cycle regulations.

In summary, our data suggest that Foxp3 as a Treg lineage-specific TF facilitates chromatin structure reorganization to establish Treg's cell identify. Since Foxp3 by itself does not function as a chromatin loop anchor to stabilize chromatin interactions, it likely cooperates with other proteins such as CTCF, cohesin, and YY1[39] to set up Treg-specific 3D genome structure during Treg development. Once the 3D genome structure is formed in mature Tregs, the contribution of Foxp3 in the maintenance of Treg 3D chromatin structure seems to be relatively minor. Instead, Foxp3 acts as a TF to activate or repress gene expression by leveraging the promoter-enhancer proximity facilitated by the Treg-specific chromatin looping structure. This study presented a model of how lineage-specific TFs function during the late or terminal stage of cell differentiation.

## Methods

### Mice

All mice were housed in a specific pathogen-free facility under a 12 h light/dark cycle, with an ambient temperature of 20–26 °C and humidity of 30–70% at the Salk Institute. Animal experiments were performed under the regulation of the Institutional Animal Care and Use Committee according to the institutional guidelines. All mice used in the present study are in the C57BL/6 genetic background. Rag1[-/-] mice purchased from the Jackson Laboratory were used for adoptive Treg cell transfer. Foxp3[Thy1.1] reporter mice[40], Foxp3[GFP] reporter mice[22], and Foxp3[GFPKO] mice[33] were used to isolate T cell populations from the thymus and Tcon and Treg cells from spleen for in-situ Hi-C, PLAC-seq, CUT&RUN, and RNA-seq experiments. Rosa-Cas9/Foxp3[Thy1.1] mice[41] were used to isolate Treg cells for CRISPR validation of the effect of Foxp3-binding sites on Helios expression.

### Generation of Foxp3 domain-swap mutant knock-in mice

The Foxp3 DSM mice were generated by CRISPR/Cas9-based genome editing[42]. Briefly, two sgRNAs containing the target sequences gRNA1(TGAAAGGGGGTCGCATATTG) and gRNA2 (AAACCACCCCGCC ACCTGGA) and Cas9 protein were used to introduce double-strand DNA breaks; a 1340 base pair (bp) single-strand DNA (ssDNA) containing the sequence encoding the three amino acids mutations (W348Q, M370T, A372P) was used to introduce Foxp3 DS mutations via homology-directed DNA repair mechanism. The gRNAs-Cas9 RNP together with ssDNA were injected into fertilized eggs derived from the Foxp3[Thy1.1] reporter mice, and then transplanted into pseudo-prepregnant recipient mice. The genomic region surrounding the target sites was amplified from genomic DNA of resultant founder progeny by PCR using the following primers: 5′-TCTGAGGAGCCCC AAGATGT-3′, 5′-CCACTCGCACAAAGCACTTG-3′. After verifying the Foxp3 DS mutations by sequencing, Foxp3 DSM mice were bred with Foxp3[Thy1.1] mice and analyzed to determine the outcomes of the Foxp3 DS mutation. Details of the ssDNA sequence are listed in Supplementary Table 1.

### T cell isolation and analysis

DN, DP, CD8SP, CD4SP, CD25[+] Treg precursor, Foxp3[lo] Treg precursor, thymic Treg cells were isolated by FACS sorting from a thymocyte suspension. Tcon and Treg cells were isolated from the spleen by pre-enrichment with EasySep Mouse CD4[+] T cell Isolation Kit (STEMCELL Technologies, Cat# 19852), and FACS sorting. The individual cell population was sorted by FACS using the following markers. DN: CD45[+]CD4[-]CD8[-]; DP:CD45[+]CD4[+]CD8[+]; CD8SP: CD45[+]CD4[-]CD8[+]; CD4SP: CD45[+]CD4[+]CD8[-]CD25[-] Foxp3-reporter[-]; CD25[+] Treg precursor: CD45.2[+]CD4[+]CD8[-]CD25[+]Foxp3-reporter[-]; Foxp3[lo] Treg precursor: CD45[+]CD4[+]CD8[-]CD25[-]Foxp3-reporter[low]; mature thymic Treg: CD45[+]CD4[+]CD8[-]CD25[+]Foxp3-reporter[+]; splenic Tcon cells: CD45[+]TCRb[+]CD4[+]CD8[-]Foxp3-reporter[-]; splenic Treg cells: CD45[+] TCRb[+]CD4[+]CD8[-]Foxp3-reporter (Thy1.1 or GFP)[+].

To analyze immune cell compositions in Foxp3 DSM mice, a single cell suspension was prepared from the spleen or lymph nodes, treated with red cell lysis buffer, and filtered through 70 μm cell strainer. For TF staining, cells were first stained for surface markers, followed by fixation and permeabilization with reagents from the Foxp3/TF Staining Buffer Set (eBioscience, 00-5521-00) and incubated with antibodies according to the manufacturer's protocol. For cytokine analysis, cells were stimulated with phorbol 12-myristate 13-acetate (PMA) (50 ng ml[-1]; Sigma), ionomycin (500 ng ml[-1]; Sigma) and GolgiStop (BD) for 5 h. Cells were incubated with cell surface antibodies on ice for 30 min, and then subjected to intracellular staining using Foxp3/TF Staining Buffer Set as described above. Samples were run on a BD FACSAria II Flow Cytometer (Becton Dickinson) and data were analyzed by FlowJo software (Tree Star). Details of antibodies, viability dye and dilutions are listed in Supplementary Table 2.

### In situ Hi-C

In situ Hi-C experiments were performed as previously described[2] using restriction enzyme MboI (NEB) with minor modifications. Briefly,

0.5 × 10⁶ cells per each developmental stage from DN to mature thymic Treg cells, and 1 × 10⁶ splenic Tcon and Treg cells were FACS-sorted and collected for individual biological replicates. Cells were resuspended in RMPI1640 medium at a concentration of $1 \times 10^6$ cells per mL, cross-linked with 1% formaldehyde for 10 min at room temperature with rotating, and subsequently lysed and digested with 100 Units of MboI/1 million cells at 37 °C for 2 h. The following steps including making of DNA ends, proximity ligation and crosslink reversal, DNA shearing and size selection, biotin pull-down and preparation for Illumine Sequencing, final amplification and purification were carried out as previously described.

## CUT and RUN
CUT&RUN experiments were performed as previously described[43] with minor modifications. To avoid the potential activation of T cells by Concanavalin-A (ConA) beads in the standard protocol, we instead used a spin-down method to collect cells as previously reported[44]. Briefly, 0.5 × 10⁶ cells per biological replicate were collected in V-bottom 96-well plate. Cells were first washed twice in Antibody Buffer (2 mM EDTA, 1x EDTA-free protease inhibitors, 0.5 mM spermidine, 1x permeabilization buffer from eBioscience™ Foxp3/TF Staining Buffer Set) by centrifugation at 800 $g$ for 6 min at 4 °C, and then incubated with normal IgG, H3K27ac antibody, or Foxp3 antibody[41] on ice for 1 h. After two washes with buffer 1 (1x EDTA-free protease inhibitors, 0.5 mM spermidine, 1x permeabilization buffer), cell pellets were incubated with pA/G-MNase (20x) enzyme (EpiCypher) in 50 µl buffer 1 at 4 °C for 1 h. Cells were washed twice in saponin buffer (0.05% (w/v) saponin, 1x EDTA-free protease inhibitors, 0.5 mM spermidine in PBS) and resuspended in 100 µl calcium buffer (2 mM CaCl₂ in buffer 2) on ice for 30 min. 100 µl 2x stop buffer (20 mM EDTA, 4 mM EGTA in saponin buffer) was then added and incubated at 37 °C for 10–20 min to release cleaved chromatin fragments. The supernatant containing chromatin fragments was collected by centrifugation and DNA was extracted using a QIAGEN MinElute kit according to manufacturer's protocol.

The CUT&RUN libraries were prepared using the NEBNext® UltraTM II DNA Library Prep Kit for Illumina® (E7645) according to manufacturer's instructions.

## PLAC-seq
PLAC experiments were performed as previously described[36]. Briefly, 20 × 10⁶ cells per biological replicate were crosslinked in 1% formaldehyde for 10 min at room temperature. After digestion with the restriction enzyme MboI, labeling DNA ends with biotin-14-dATP, and proximity ligation of DNA ends, the resultant chromatin was sonicated to 200–600 bp using a Covaris E229 sonicator for 10 min. Sonicated chromatin was pre-cleared with Protein A + G magnetic beads followed by overnight chromatin immunoprecipitations with 5 µg Foxp3 antibody[41]. Libraries were prepared using NuGen Ovation Ultralow Library System V2 kit according to the manufacturer's instructions and sequenced by an Illumina HiSeq2500 sequencer.

## RNA-seq
1 × 10⁴ cells were FACS-sorted into TRIzol RNA isolation reagent (Invitrogen) and RNA was isolated according to the protocol. RNA concentration and integrity was determined by Bioanalyzer using RNA 6000 Pico Kit (Agilent). RNA-seq libraries were constructed using Illumina TruSeq Stranded mRNA kit (Illumina) following manufacturer's instructions.

## Serum immunoglobulin ELISA
Serum IgG1, IgG2c, IgM, and IgA concentrations were measured by ELISA using the SBA Clono-typing System (Southern Biotech).

## Histology
For histology analysis, lung, liver, small intestine, and salivary gland tissues were fixed in 10% neutral buffer formalin, paraffin-embedded, sectioned, and stained with hematoxylin and eosin by Pacific Pathology (San Diego, CA).

## In vitro culture of Treg cells
Treg cells were isolated from spleen of the Foxp3^Thy1.1 reporter mice by staining with PE-labeled Thy1.1 antibody followed by enrichment with Anti-PE magnetic beads (Miltenyi, Cat# 130-048-801). Purity of the Treg cells was confirmed to be over 95% by FACS. Treg cells were activated by plate-bound anti-CD3 and anti-CD28 antibodies in complete RPMI1640 medium containing 10% FBS, 100 U/ml Penicillin/Streptomycin, 1x GlutaMax, 1 mM HEPES, 1 mM sodium pyruvate, 1x NEAA, 55 mM 2-mercaptoethanol, 10 mg/ml gentamycin, and IL-2 at 500 U/ml.

## CRISPR knockout of Foxp3 binding sites in *Ikzf2* locus
Cloning of sgRNAs into the pSIRG-NGFR vector, and retrovirus production in HEK293T cells were performed as previously described[45]. Details of sgRNA sequence are listed in Supplementary Table 3.

To test the effects of Foxp3 binding sites on Helios expression in vitro, Treg cells isolated from Rosa-Cas9/Foxp3^Thy1.1 mice were transduced with retrovirus carrying the sgRNAs targeting P1 region, P6 region, or control non-targeting sgRNAs. Cells were stained and analyzed for Helios expression by FACS 5 days after transduction.

## In vivo adoptive Treg cell transfer
To test the effect of Foxp3 binding sites in the *Ikzf2* locus on Helios expression in vivo, Treg cells from Rosa-Cas9 mice were first transduced with control sgRNAs, sgRNAs targeting P1 or P6 region on day 1, and then sorted by FACS on day 3 to enrich Treg cells transduced with sgRNAs (CD4⁺Foxp3^Thy1.1+NGFR⁺). CD45.2⁺ Treg cells transduced with control sgRNAs were mixed with CD45.1⁺CD45.2⁺ Treg cells transduced with sgRNAs targeting P1 or P6 region respectively at a ratio of 50:50 into RAG1 KO mice. 2 weeks after transfer, T cells in the spleen were harvested for analysis of the expression of Helios.

## In-situ Hi-C and PLAC-seq data analysis
Hi-C and PLAC-seq data were aligned to the mm9 reference genome using BWA-MEM[46]. Reads were filtered (MAPQ > = 30) and paired using a previously described pipeline[47]. PCR duplicate reads were removed using Picard. Contact matrices were generated and normalized using the iterative correction method[48]. Enriched contacts in the PLAC-seq data were identified using HICCUPs[49].

To detect differential chromatin interactions between experiments, we calculated contact frequencies in 25 kb bins for all interactions genome wide separated by less than 1 Mb. Differential interacting regions were called using edgeR[50] with Benjamini-Hochberg correction for multiple testing.

TF motif enrichment in Foxp3-dependent chromatin interaction sites was performed by first identifying differential chromatin interactions between wild-type (Foxp3GFP⁺) and Foxp3 knockout (KIKO GFP⁺) Tregs at a resolution of 25 kb. Differential interacting regions were overlapped with DHS in mouse Tregs from ENCODE (accession ENCFF566TDU). Motif enrichment over these DHS sites found in differential interaction regions was performed using Homer[51] using the collection of known vertebrate motifs from the JASPAR database.

## RNA-seq data analysis
RNA-seq data were aligned by using STAR[46] to the mm9 reference genome. PCR duplicates were removed, and read counts were quantified over GENCODE genes (vM1) using HTSeq and subject to RPKM

normalization. Differentially expressed genes were identified using edgeR[50]. GO analysis was performed using TopGO[52].

## CUT&RUN data analysis

CUT&RUN data was aligned to the mm9 reference genome using BWA-MEM[46]. Peaks were called using MACS2[53]. Peaks calls from the Foxp3 experiments together to create a union set of peaks. Using this merged peak set, we then identified differential Foxp3 sites using edgeR[50].

## Reporting summary

Further information on research design is available in the Nature Portfolio Reporting Summary linked to this article.

## Data availability

The sequencing data from the RNA-seq, ChIP-seq, CUT&RUN, Hi-C, and PLAC-seq experiments have been deposited in Gene Expression Omnibus (GEO) with the following accession ID: GSE217147. [https://www.ncbi.nlm.nih.gov/geo/query/acc.cgi]. Source data are provided with this paper.

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

## Acknowledgements

We would like to thank T. Hua, N. Tessler, and C. Gordon for mouse colony management, C. O'Connor for assistance in flow cytometry, Y. Dayn for assistance in generating the Foxp3 DSM mutant mice. Z.L. was supported by the National Natural Science Foundation of China (32370937), a NOMIS Fellowship. D.S.L. was supported by the National Research Foundation (NRF) of Korea awards NRF2021R1C1C100679813, NRF2022M3A9D301684812, and NRF2022M3H9A108101113. Y.Z. was supported by the NOMIS Foundation, the Crohn's and Colitis Foundation, the Sol Goldman Trust, and National Institutes of Health (R01-AI107027, R01-AI1511123, R21-AI154919, and S10-OD023689). J.R.D. was supported by grants from National Institutes of Health (DP5-OD023071 and U01-CA260700). This study was supported by Shanghai Immune Therapy Institute, and was also supported by National Cancer Institute funded Salk Institute Cancer Center Core Facilities (P30-CA014195).

## Author contributions

Conceptualization: Z.L., D.S.L., Y.Z. and J.R.D. Methodology: Z.L. and D.S.L. Investigation: Z.L., D.S.L. and Y.L. Resources: Y.Z. and J.R.D. Formal analysis: Z.L., D.S.L. and J.R.D. Data Curation: Z.L. D.S.L. and J.R.D. Supervision: Y.Z. and J.R.D. Funding acquisition: Y.Z. and J.R.D Writing – original draft preparation: Z.L., D.S.L., Y.Z. and J.R.D. Writing – review and editing: Z.L., D.S.L., Y.Z. and J.R.D.

## Competing interests

The authors declare no competing interests.
