## [Peer Review File · Nature Communications]

Foxp3 Orchestrates Reorganization of Chromatin Architecture to Establish Regulatory T Cell IdentityREVIEWER COMMENTS

Reviewer #1 (Remarks to the Author):

In this manuscript, Liu et al. performed HiC and Foxp3 HiChIP experiments to address how the 3D chromatin structure of Treg cells is established during their development in the thymus, how it influences their gene expression, and whether and how Foxp3 contributes to the 3D chromatin structure in Treg cells. They showed that; 1) chromatin interactions are reorganized during thymic differentiation of Treg cells; 2) a fraction of Treg or Tconv-specific chromatin interactions are associated with Treg or Tconv-specific gene expression; 3) Foxp3 binding sites are enriched at Treg-specific chromatin loop anchors; 4) a fraction of the Treg-specific chromatin interactions are lost in Foxp3-null Treg-like cells (Treg wannabes) but retained in Treg-like cells expressing the domain-swap mutant of Foxp3; 5) promoters of some Treg-specific or Tconv-specific genes overlap with Foxp3-associated chromatin interactions; 6) deletion of Foxp3 in mature Treg cells failed to alter Treg-specific chromatin interactions. Based on these findings, the authors propose that Foxp3 facilitates the formation but not maintenance of Treg-specific chromatin structure and thereby regulates Treg-specific gene expression.

A previous study (ref #27) conducted H3K27Ac and Foxp3 HiChIP in Treg and Tconv and proposed that Foxp3 is associated with, and facilitates formation of, enhancer-promoter loops of many genes including Foxp3-dependent Treg-specific genes. The present study provides not only further evidence for the essential role of Foxp3 in the formation of higher-order chromatin structure in Treg cells but also novel insights into when and how this structure is established and maintained in Treg cells. I am in general supportive of publication of this manuscript in Nature Communications, but there are some issues that need to be addressed prior to consideration for publication.

Major issues

1. Fig. 2: Although the authors showed in Fig. 1 that 3D chromatin structure is reorganized during thymic differentiation of Treg cells, it is not clear at what stage the Treg and Tconv-specific chromatin interactions identified in Fig. 2A are established. The authors should analyze changes of these interactions during thymic differentiation.
2. Fig. 2D, F, G: It is not clear from these analyses whether increased chromatin interactions are associated with increased gene expression. The authors should separate the differential chromatin interaction anchors into Treg- and Tconv-specific ones and examine whether Treg-specific and Tconv-specific interactions are correlated with expression of Treg-specific and Tconv-specific genes, respectively.
3. Fig. 3: Although the authors show a fraction of Treg-specific chromatin interactions are lost in Treg wannabes, it is not clear whether Tconv-specific as well as common chromatin interactions are also affected. It is conceivable that Foxp3 may interfere with the formation of Tconv-specific chromatin interactions in Treg cells. If so, such interactions would be increased in Treg wannabes. In Fig. 3C, the authors should include log₂FC values for the Tconv-specific and common chromatin interactions and make comparisons among the three groups.
4. Fig. 3: The authors should address whether the differences in chromatin interactions between Treg and Treg wannabes are associated with differences in gene expression.
5. Fig. 6E: The authors should also address whether genes with Foxp3-associated chromatin interactions are expressed in a Foxp3-dependent manner.
6. Fig. 3: It is not described whether Foxp3 wannabes were isolated from Foxp3 KIKO/Y hemizygous male or KIKO/+ heterozygous female mice. If male mice were used, how can the authors rule out the possibility that the observed changes in chromatin interactions are indirect consequences of the inflammatory condition in these sick mice?

7. Because the data produced in this study are valuable resources for the research community, the authors should provide differential chromatin interactions between Treg and Tconv, Foxp3-associated chromatin interactions, differentially expressed genes between WT and Foxp3 mutant (null and DSM) Treg cells, differential Foxp3 binding between WT and DSM mutant Treg cells as supplementary tables.

8. I cannot understand the message of Figure 7 in the context of this manuscript. I think that this figure is rather distractive than helpful.

Other points

9. Fig. 1D is called out as Fig. 1E in the main text and figure legends, and vice versa.

10. Typos should be corrected.

- Line #114: Figure 1B should be Figure 2B.
- Line #123: FDR > 1% should be FDR < 1%.
- Line #128: Figure 1D should be Figure 2D.
- Line #243: Naïve T cells should be Tcon.
- Fig. 5D: X-axis should be log₂ FC (DSM/WT) not (WT/DSM).

11. The number of Treg-specific chromatin interactions is not consistent. In line #111, it says 1959, in line #164 1569, in line #230 1445. Which one is correct?

12. Fig. 6H: Dots should not be connected by lines unless connected dots are paired observations, which should not be the case. Accordingly, unpaired t-test should be used.

Reviewer #3 (Remarks to the Author):

Liu et al. present a well-executed research project that investigates the chromatin conformation changes in mouse T cell development, focusing on the effects of foxp3 deletion and foxp3 mutants on Treg cell development. By generating a comprehensive collection of Hi-C data from various developmental stages, the study provides novel insights into the role of Foxp3 in DNA looping, with results that go in contrast with prior evidence based on Foxp3 structure, 4C, and HiChIP data. The distinct findings from this study, which employs foxp3 mutants rather than knockout models, contribute significant knowledge to the field.

Main comments:

1. The authors conduct differential loop analysis by comparing all 25kb bins within a 1Mb distance, leading to a high number of comparisons that may reduce statistical power. It is recommended that they consider testing only bins containing a loop identified by HICCUPS to test if this leads to improved power.
2. The study refers to foxp3 knockout Tregs as "wannabe" Tregs. However, the extent of their similarity to actual Tregs remains unclear, as they lack foxp3. Similar concerns apply to the domain swap mutant (DSM). It is essential to elucidate the functions of foxp3 affected by the DSM mutant, if effects have been previously described, they should be explained appropriately in the results and discussion sections to make it easier to understand. With the current explanation, it seems that the statement in the discussion of the article stating that "Foxp3 is required to form the 3d genome structure specific to Tregs, but once the 3d genome structure is formed in mature Tregs, it is not required for its maintenance" seems a bit of a stretch. After reading the article cited in ref 35, this statement makes more sense, but more information should be provided in the manuscript.
3. While the authors isolate T cells at various developmental stages, most comparisons focus on Tcon and Treg cells. Additional analyses involving other stages could provide a more comprehensive understanding of the chromatin conformation changes during T cell development.

Minor comments:

1. The introduction discusses Treg cell lineage specification as the final stage of T cell development. To provide context for readers unfamiliar with the subject, a brief description of earlier T cell development stages would be helpful.
2. To benefit readers who may be less familiar with the in situ Hi-C and PLAC-seq technique employed in the study, the authors should provide more information about the method.
3. In the second section of the results, the authors mention the number of upregulated interactions without specifying the FDR threshold used. Please provide this information.

Response to reviewers

We would like to thank the reviewers for their constructive criticisms and helpful suggestions to improve our manuscript. We have devoted our efforts to addressing reviewers' concerns by refining our analysis and providing more clear interpretations in the revised manuscript. It is our sincere hope that the reviewers and the editor find the revised manuscript acceptable for publication.

Reviewer #1 (Remarks to the Author):

In this manuscript, Liu et al. performed HiC and Foxp3 HiChIP experiments to address how the 3D chromatin structure of Treg cells is established during their development in the thymus, how it influences their gene expression, and whether and how Foxp3 contributes to the 3D chromatin structure in Treg cells. They showed that; 1) chromatin interactions are reorganized during thymic differentiation of Treg cells; 2) a fraction of Treg or Tconv-specific chromatin interactions are associated with Treg or Tconv-specific gene expression; 3) Foxp3 binding sites are enriched at Treg-specific chromatin loop anchors; 4) a fraction of the Treg-specific chromatin interactions are lost in Foxp3-null Treg-like cells (Treg wannabes) but retained in Treg-like cells expressing the domain-swap mutant of Foxp3; 5) promoters of some Treg-specific or Tconv-specific genes overlap with Foxp3-associated chromatin interactions; 6) deletion of Foxp3 in mature Treg cells failed to alter Treg-specific chromatin interactions. Based on these findings, the authors propose that Foxp3 facilitates the formation but not maintenance of Treg-specific chromatin structure and thereby regulates Treg-specific gene expression.

A previous study (ref #27) conducted H3K27Ac and Foxp3 HiChIP in Treg and Tconv and proposed that Foxp3 is associated with, and facilitates formation of, enhancer-promoter loops of many genes including Foxp3-dependent Treg-specific genes. The present study provides not only further evidence for the essential role of Foxp3 in the formation of higher-order chromatin structure in Treg cells but also novel insights into when and how this structure is established and maintained in Treg cells. I am in general supportive of publication of this manuscript in Nature Communications, but there are some issues that need to be addressed prior to consideration for publication.

Major issues

1. Fig. 2: Although the authors showed in Fig. 1 that 3D chromatin structure is reorganized during thymic differentiation of Treg cells, it is not clear at what stage the Treg and Tconv-specific chromatin interactions identified in Fig. 2A are established. The authors should analyze changes of these interactions during thymic differentiation.

We thank the reviewer for this suggestion. We followed the reviewer's suggestion and re-analyzed the chromatin interactions during thymic differentiation. Treg-specific chromatin interactions emerge most dramatically upon Foxp3 expression in the Foxp3+CD25- population, with some chromatin interactions being further strengthened upon additional CD25 expression in the Foxp3+CD25+ population. On the contrary, Tcon-specific chromatin interactions are largely lost upon expression of Foxp3 (Foxp3+CD25-), and most do not show a further reduction upon expression of CD25

(Foxp3+CD25+). These results have now been incorporated in the main text as part of Supplementary Fig. 1C.

2. Fig. 2D, F, G: It is not clear from these analyses whether increased chromatin interactions are associated with increased gene expression. The authors should separate the differential chromatin interaction anchors into Treg- and Tconv-specific ones and examine whether Treg-specific and Tconv-specific interactions are correlated with expression of Treg-specific and Tconv-specific genes, respectively.

We appreciate the reviewer's suggestion. We have compared the patterns of Treg or Tcon specific chromatin interactions with Treg or Tcon differentially expressed genes. Treg-specific chromatin interactions are strongly associated with Treg specifically expressed genes, and that Tcon specific interactions are strongly associated with Tcon specifically expressed genes as illustrated by the figure below. This figure has now been incorporated as Fig. 2D (replacing the old Fig. 2D panel).

3. Fig. 3: Although the authors show a fraction of Treg-specific chromatin interactions are lost in Treg wannabes, it is not clear whether Tconv-specific as well as common chromatin interactions are also affected. It is conceivable that Foxp3 may interfere with the formation of Tconv-specific chromatin interactions in Treg cells. If so, such interactions would be increased in Treg wannabes. In Fig. 3C, the authors should

include log₂FC values for the Tconv-specific and common chromatin interactions and make comparisons among the three groups.

We appreciate the reviewer's suggestion. We performed the analysis suggested by the reviewer (shown below). Indeed, we see that the Tcon-specific chromatin interactions increased in the KIKO Treg cells compared to wild-type as predicted by the reviewer, suggesting that Tcon-specific interactions are negatively affected by Foxp3 in WT Tregs. The panel below is now included as Fig. 3C.

4. Fig. 3: The authors should address whether the differences in chromatin interactions between Treg and Treg wannabes are associated with differences in gene expression.

We thank the reviewer for this comment. We compared the differentially expressed genes between Foxp3-KIKO and WT Tregs. We made two comparisons, one with all differential interactions between Tcon and Treg cells, and the other only with differential chromatin interactions between Foxp3-KIKO and WT Tregs. We were able to see that in both cases, the differentially expressed genes were enriched at differential chromatin interactions as illustrated by the figure below. This analysis has now been incorporated as part of Fig. 3.

5. Fig. 6E: The authors should also address whether genes with FoXP3-associated chromatin interactions are expressed in a FoXP3-dependent manner.

We thank the reviewer for this comment. We compared the FoXP3-associated chromatin interactions as measured by PLAC-seq to genes expressed in a FoXP3-dependent manner, as defined as being differentially expressed in the FoXP3-KIKO vs. WT Treg cells. Indeed, we observed that FoXP3-associated interactions are enriched for FoXP3-dependent genes. The enrichment, while significant, is not dramatic. We believe this may be related to the fact that only a minority of genes that are associated with FoXP3-associated chromatin interactions are differentially expressed between Tcon and Treg cells (Fig. 6E). This may suggest that only a minority of the FoXP3-associated chromatin interactions are truly consequential for regulatory enhancer-promoter type interactions.

6. Fig. 3: It is not described whether Foxp3 wannabes were isolated from Foxp3 KIKO/Y hemizygous male or KIKO/+ heterozygous female mice. If male mice were used, how can the authors rule out the possibility that the observed changes in chromatin interactions are indirect consequences of the inflammatory condition in these sick mice?

We thank the reviewer for raising this concern. Indeed, we isolated Foxp3 wannabes from KIKO/WT heterozygous female mice for all the experiments Foxp3 wannabes involved to avoid the situation as the reviewer stated. In the KIKO/WT female mice, about half of the Tregs are WT Tregs, which is sufficient to maintain immune system homeostasis. Therefore, the differences we observed between Treg and Treg wannabes were cell intrinsic, and not due to inflammatory conditions.

We added detailed descriptions as follows in the main text:

“Hi-C experiments were performed with GFP·Foxp3⁻ Treg- “wannabe” cells from healthy heterozygous female Foxp3^{GFP-KIKO/WT} mice and control GFP·Foxp3⁺ WT Treg cells from Foxp3^{GFP/WT}Foxp3^{GFP} mice.”

7. Because the data produced in this study are valuable resources for the research community, the authors should provide differential chromatin interactions between Treg and Tconv, Foxp3-associated chromatin interactions, differentially expressed genes

between WT and Foxp3 mutant (null and DSM) Treg cells, differential Foxp3 binding between WT and DSM mutant Treg cells as supplementary tables.

We have provided the requested data in supplementary tables in the revised manuscript (as Supplementary Table 2-6).

8. I cannot understand the message of Figure 7 in the context of this manuscript. I think that this figure is rather distractive than helpful.

We understand the reviewer's concern and would like to provide a better explanation for the significance of Fig. 7. The purpose of Fig. 7 was to better explore the downstream regulatory consequences of Foxp3 expression in Treg cells. For example, in analyzing the Foxp3-KIKO vs. WT Treg chromatin interactions, we observed that chromatin interactions that were more strongly lost in Foxp3-KIKO Tregs showed different motifs (including Forkhead motifs) compared with chromatin interactions that were retained in Foxp3-KIKO Tregs (such as ETS motifs and CTCF). This suggests to us that Foxp3 cooperates with other transcriptional regulators to affect the final Treg gene expression landscape. In light of this, we were interested in exploring whether Foxp3 controlled any particular aspect of gene expression during Treg commitment, and we indeed observed that this was the case. We have edited the text to help better clarify the rationale of this figure and its relevance to the rest of the manuscript.

Other points

9. Fig. 1D is called out as Fig. 1E in the main text and figure legends, and vice versa.

We thank the reviewer for pointing this out. We have switched panels D and E in the figure so the legend and callouts now match.

10. Typos should be corrected.

- Line #114: Figure 1B should be Figure 2B.
- Line #123: FDR > 1% should be FDR < 1%.
- Line #128: Figure 1D should be Figure 2D.
- Line #243: Naïve T cells should be Tcon.
- Fig. 5D: X-axis should be log₂ FC (DSM/WT) not (WT/DSM).

We thank the reviewer for pointing these out. We have corrected all of the above errors.

11. The number of Treg-specific chromatin interactions is not consistent. In line #111, it says 1959, in line #164 1569, in line #230 1445. Which one is correct?

We thank the reviewer for pointing out the inconsistency. This was an oversight on our part related to the methods for calling differential chromatin interactions. The tool we use (edgeR) automatically filters out some of the data if the observed counts do not meet an arbitrary threshold. We used these default conditions, but in some cases this causes the tool to filter out some of the Treg-specific chromatin interactions. We have

corrected this so that in each case we are considering all 1959 Treg-specific chromatin interactions. We have updated the main text to reflect this as well as in the panels in Fig. 3 and Fig. 5.

12. Fig. 6H: Dots should not be connected by lines unless connected dots are paired observations, which should not be the case. Accordingly, unpaired t-test should be used.

We designed the experiment to co-transferred allelically marked sgCtrl Tregs together with sgP1 or sgP6 Tregs into the same host and analyzed the expression of Helios in a pairwise way. Therefore, we think using paired t-test is suitable for this analysis.

Reviewer #3 (Remarks to the Author):

Liu et al. present a well-executed research project that investigates the chromatin conformation changes in mouse T cell development, focusing on the effects of *foxp3* deletion and *foxp3* mutants on Treg cell development. By generating a comprehensive collection of Hi-C data from various developmental stages, the study provides novel insights into the role of Foxp3 in DNA looping, with results that go in contrast with prior evidence based on Foxp3 structure, 4C, and HiChIP data. The distinct findings from this study, which employs *foxp3* mutants rather than knockout models, contribute significant knowledge to the field.

Main comments:

1. The authors conduct differential loop analysis by comparing all 25kb bins within a 1Mb distance, leading to a high number of comparisons that may reduce statistical power. It is recommended that they consider testing only bins containing a loop identified by HICCUPS to test if this leads to improved power.

We thank the reviewer for the insightful comment. In fact, the analytical approach recommended by the reviewer was the same strategy we pursued in our initial analysis during this study. However, in examining the chromatin interaction maps and CUT&RUN profiles, we recognized that the chromatin conformation differences between Treg and Tcon cells were not limited to loops. In addition, Foxp3 binding sites show only partial overlap with CTCF binding sites, which is a major driver of loop formation. As a result, we expanded the analysis to consider all 25kb bins within 1Mb as described in the current manuscript.

In light of the reviewer's suggestion, we directly compared the strategy of first calling loops and then identifying differential chromatin interactions with the approach we outlined in the manuscript. We identified a total of 12,441 loops in Treg and Tcon. At a false discovery rate (FDR) of less than 1%, the number of differentially interacting loops stood at 984 (7.9%). We also identified 1,577 differential loop anchors. Of these 1,577 differential loop anchors, 53.5% (844) overlapped with differential loop anchors we identified using our genome-wide 25kb bin strategy, showing good agreement between

the two approaches. However, our genome-wide approach identified a total of 4202 differential anchors for chromatin interactions, suggesting that the strategy of calling loops first fails to identify most (79.9%) of the differences in 3D genome structure we identify using our genome-wide 25kb bin approach.

Hence we remain confident that the broad-based analysis presented in the current manuscript provides a more nuanced and accurate reflection of the biology of Treg development. While we acknowledge the value of focusing on chromatin loops, we will stay with our current approach of considering a wider range of interactions in order to capture the complexity inherent in this process and provide a comprehensive view of our findings.

Thank you once again for your thoughtful and constructive feedback.

2. The study refers to *foxp3* knockout Tregs as "wannabe" Tregs. However, the extent of their similarity to actual Tregs remains unclear, as they lack *foxp3*. Similar concerns apply to the domain swap mutant (DSM). It is essential to elucidate the functions of *foxp3* affected by the DSM mutant, if effects have been previously described, they should be explained appropriately in the results and discussion sections to make it easier to understand. With the current explanation, it seems that the statement in the discussion of the article stating that "Foxp3 is required to form the 3d genome structure specific to Tregs, but once the 3d genome structure is formed in mature Tregs, it is not required for its maintenance" seems a bit of a stretch. After reading the article cited in ref 35, this statement makes more sense, but more information should be provided in the manuscript.

We appreciate the reviewer for this comment. We will explain and rephrase this point more appropriately in the revised version as follows:

For Treg "wannabe" cells

"As reported previously, Treg "wannabe" cells receive similar TCR signals and developmental cues like WT Tregs, express the majority of Treg signature genes, and might represent Treg precursors, as reflected by mRNA transcriptional activity at the *Foxp3* locus but lack *Foxp3* protein expression and suppressive function."

For *Foxp3* DSM cells:

"Of note, three amino acid mutations (W348Q, M370T and A372P) in the domain-swap interface that disrupt domain-swap dimer formation diminish Treg suppressive activity without compromising *Foxp3* DNA binding in vitro, indicative of a direct role of *Foxp3* in regulating Treg function by modulating Treg cell 3-D chromatin structure."

As for "Foxp3 is required to form the 3D genome structure specific to Tregs, but once the 3D genome structure is formed in mature Tregs, it is not required for its

maintenance”, we soften our statement a notch in the discussion as follows to make it more balanced:

“This data indicates that although Foxp3 plays an essential role in the formation of Treg-specific chromatin interactions, it has only marginal effects on the maintenance of their 3D chromatin structure”

“Once the 3D genome structure is formed in mature Tregs, the contribution of Foxp3 in the maintenance of Treg 3D chromatin structure seems to be relatively minor.”

3. While the authors isolate T cells at various developmental stages, most comparisons focus on Tcon and Treg cells. Additional analyses involving other stages could provide a more comprehensive understanding of the chromatin conformation changes during T cell development.

We appreciate the reviewer’s suggestion. A similar point was also raised by reviewer #1 (comment #5). To address this, we re-analyzed the chromatin interactions during thymic differentiation at sites we initially identified as Treg or Tcon-specific chromatin interactions. Treg-specific chromatin interactions emerge most dramatically upon Foxp3 expression (Foxp3+CD25-), with some chromatin interactions being further strengthened upon additional CD25 expression (Foxp3+CD25+). On the contrary, Tcon-specific chromatin interactions are largely lost upon expression of Foxp3 (Foxp3+CD25-) and most do not show a further reduction upon expression of CD25 (Foxp3+CD25+). These results have now been incorporated in the main text as part of Supplementary Fig. 1C.

We also agree with the reviewer that there is likely more to explore in the data from different T-cell developmental stages. However, given that the rest of the manuscript focuses on mature Treg chromatin interaction and gene expression, we believe that an in-depth exploration of 3D genome changes in developing T cells would be a distraction from the main points of the study.

We also recognize that these T cell development datasets are valuable to the community. As a result, we have made our data publicly available, including in browsable forms such as *.hic files, so that interested readers can easily download and visualize the data for themselves.

Minor comments:

1. The introduction discusses Treg cell lineage specification as the final stage of T cell development. To provide context for readers unfamiliar with the subject, a brief description of earlier T cell development stages would be helpful.

We will include a brief introduction about “Treg cell lineage specification as the final stage of T cell development” at the beginning of the second paragraph of the introduction as follows:

T cell development in the thymus occurs through several stages distinguished by the expression of cell surface markers CD4 and CD8. Early T cell precursors are CD4/CD8 double-negative (DN), which differentiate into the CD4/CD8 double-positive (DP) intermediate population, and then the mature CD4 single-positive (SP) or CD8 SP populations (Figure 1A). Regulatory T cells (Treg) are derived in the thymus from CD4 SP cells, through two distinct developmental programs involving either CD25⁺Foxp3⁻ or CD25⁻Foxp3⁺ Treg cell precursors, both of which develop into CD25⁺Foxp3⁺ mature Treg cells to maintain immune tolerance and homeostasis.

2. To benefit readers who may be less familiar with the in situ Hi-C and PLAC-seq technique employed in the study, the authors should provide more information about the method.

We thank the reviewer for this comment. We have added text to the main text to better familiarize readers with the methods. Specifically, we write:

For in situ Hi-C:

“*In situ* Hi-C is a genome-wide variant of chromatin conformation assay that provides an all-to-all, high resolution view of chromatin interactions in the genome.”

For PLAC-seq:

“PLAC-seq is an assay that combines Hi-C experiments with ChIP, such that chromatin interactions associated with a factor of interest can be studied in higher resolution.”

3. In the second section of the results, the authors mention the number of upregulated interactions without specifying the FDR threshold used. Please provide this information.

We thank the reviewer for pointing this out. The FDR used for the analysis of differential chromatin interactions in Treg vs. Tcon cells was 1%. We have updated the main text to include this.

REVIEWERS' COMMENTS

Reviewer #1 (Remarks to the Author):

The additional analyses performed by the authors have satisfactorily addressed all the concerns raised in my previous review. I strongly recommend this manuscript for publication in Nature Communications.

Reviewer #3 (Remarks to the Author):

The authors responded to all the comments provided previously in a satisfactory manner and I have no further comments.